# Barcoded multiple displacement amplification for high coverage sequencing in spatial genomics

Jinhyun Kim[1], Sungsik Kim[2], Huiran Yeom[3], Seo Woo Song[4], Kyoungseob Shin[1], Sangwook Bae[5], Han Suk Ryu[6,7], Ji Young Kim[8], Ahyoun Choi[2], Sumin Lee[1,15], Taehoon Ryu[9], Yeongjae Choi[10], Hamin Kim[2], Okju Kim[9], Yushin Jung[9], Namphil Kim[1], Wonshik Han[6,8,11], Han-Byoel Lee[6,8,11] ✉, Amos C. Lee[12,15] ✉ & Sunghoon Kwon[1,2,8,12,13,14] ✉

Determining mutational landscapes in a spatial context is essential for understanding genetically heterogeneous cell microniches. Current approaches, such as Multiple Displacement Amplification (MDA), offer high genome coverage but limited multiplexing, which hinders large-scale spatial genomic studies. Here, we introduce barcoded MDA (bMDA), a technique that achieves high-coverage genomic analysis of low-input DNA while enhancing the multiplexing capabilities. By incorporating cell barcodes during MDA, bMDA streamlines library preparation in one pot, thereby overcoming a key bottleneck in spatial genomics. We apply bMDA to the integrative spatial analysis of triple-negative breast cancer tissues by examining copy number alterations, single nucleotide variations, structural variations, and kataegis signatures for each spatial microniche. This enables the assessment of subclonal evolutionary relationships within a spatial context. Therefore, bMDA has emerged as a scalable technology with the potential to advance the field of spatial genomics significantly.

Integrating genomic aberrations in each spatial microniche in the tumor microenvironment provides thorough insights into tumor development[1]. Despite its importance in understanding cancer, the integrated analysis of genomic aberrations, such as copy number alterations (CNA)[2], single nucleotide variants (SNV)[3,4], structural variations (SV)[5], and kataegis[6], poses technical challenges within the spatial context. Achieving a higher spatial resolution requires a scalable genome amplification technology that can simultaneously process multiple spatial microniches[7], whereas integrative genomics requires high genome coverage of each spatial microniche.

[1]Department of Electrical and Computer Engineering, Seoul National University, Seoul 08826, Republic of Korea. [2]Interdisciplinary Program in Bioengineering, Seoul National University, Seoul 08826, Republic of Korea. [3]Division of Data Science, College of Information and Communication Technology, The University of Suwon, Hwaseong 18323, Republic of Korea. [4]Basic Science and Engineering Initiative, Children's Heart Center, Stanford University, Stanford, CA, USA. [5]Renal Division and Division of Engineering in Medicine, Department of Medicine, Brigham and Women's Hospital, Harvard Medical School, Boston, MA, USA. [6]Cancer Research Institute, Seoul National University, Seoul 03080, Republic of Korea. [7]Department of Pathology, Seoul National University College of Medicine, Seoul 03080, Republic of Korea. [8]Biomedical Research Institute, Seoul National University Hospital, Seoul 03080, Republic of Korea. [9]ATG Lifetech Inc., Seoul 08507, Republic of Korea. [10]School of Materials Science and Engineering, Gwangju Institute of Science and Technology (GIST), Gwangju 61005, Republic of Korea. [11]Department of Surgery, Seoul National University College of Medicine, Seoul 03080, Republic of Korea. [12]Bio-MAX Institute, Seoul National University, Seoul 08826, Republic of Korea. [13]Inter-University Semiconductor Research Center, Seoul National University, Seoul 08826, Republic of Korea. [14]Institutes of Entrepreneurial BioConvergence, Seoul National University, Seoul 08826, Republic of Korea. [15]Present address: Meteor Biotech, Co. Ltd., Seoul 08826, Republic of Korea. ✉e-mail: hblee80@gmail.com; amoslee89@gmail.com; skwon@snu.ac.kr

There is high demand for scalable genome amplification technology that can process numerous spatial microniches with significantly reduced costs to achieve integrative genome analysis across the entire tumor space. For example, the majority of spatial omics technologies that profile tumor architectures, according to genomics[1,8–10], transcriptomics[11–14], epigenomics[15,16], and multi-omics[12], utilize in situ barcoding as a core strategy to become scalable. By barcoding biomolecules with region-specific DNA sequences, the barcoded products can be pooled to perform subsequent library preparation in one pot, thereby enabling spatial tumor assays with reduced cost and labor. However, these technologies are either not applicable to genome analysis or have low genome coverage, which allows the analysis of only CNA out of several other complex genomic aberrations[17–19]. This has resulted in the need for a scalable, high-coverage genome amplification technology for integrative spatial genomics; however, this remains technically challenging.

Multiple displacement amplification (MDA) is widely preferred for amplifying whole genomes with high genome coverage[7,20–25], which is a prerequisite for the integrative analysis of genomic aberrations. Unlike other polymerase chain reaction (PCR)-based genome amplification technologies that can amplify only specific genomic regions with fixed DNA sequences, MDA uses random hexamers as primers to broadly initiate amplification across the whole genome. Another breakthrough enabled by MDA was the use of a primer concentration ~100 times higher than that used in conventional PCR to enable phi29 DNA polymerase-mediated uniform and exponential amplification of displaced DNA strands. These characteristics of MDA, which facilitate the amplification of an arbitrary genome with high coverage, have contributed substantially to several scientific discoveries, such as the detection of SNV and kataegis from single cells[26,27]. Investigating the SNV for each spatial microniche became feasible with MDA[26,28] and resulted in answers to long-held questions on the evolution of a tumor over time[21,29,30], metastasis of cancer[31], and progression of pre-malignant cancers to invasive malignancies[4]. However, analyzing the landscape of genomic aberrations in a large set of spatial microniches using MDA requires a large budget for the preparation of next-generation sequencing libraries from MDA products. Although the cost of NGS is decreasing rapidly, MDA and library preparation can be a significant portion of the overall cost of whole-genome sequencing. Therefore, applying a barcoding strategy in spatial genomics can increase the scalability of MDA. However, in contrast to other scalable single-cell omics technologies that usually utilize PCR-based barcoding strategies with barcoded primers, where barcode addition negligibly affects the amplification reaction, modification of the MDA primers significantly inhibited the MDA reaction because of the atypically high primer concentration (Supplementary Fig. 1). Therefore, realizing barcode-based scalable MDA is technically challenging, and the high cost and low-throughput nature of the MDA workflow considerably hamper the analysis of a large number of spatial microniches[17–19], even though MDA has innovative potential in the life sciences. This inhibition-free barcoded MDA method will significantly advance our understanding of heterogeneous cellular systems by allowing large-scale analysis of spatial microniches in a cost-effective and high-throughput manner.

Here, we demonstrate barcoded MDA (bMDA) to realize scalable and high-coverage genome analysis, which would enable an in-depth analysis of the spatial genomic landscape (Fig. 1). In bMDA, cell barcodes are incorporated into the MDA products by replacing the conventional MDA primer with a barcoded primer, enabling sample pooling before the subsequent library preparation step. The main technical hurdle of the MDA reaction inhibition arising from primer modification is explored to identify the key factors for realizing bMDA (Supplementary Fig. 2). By reshaping the conventional workflow of MDA-prep-and-pool (Fig. 1b) into MDA-pool-and-prep (Fig. 1c), our approach allows one-pot library preparation, which reduces the complicated and labor-intensive library preparation steps to $1/N$, where $N$ is the number of samples to be simultaneously analyzed. We demonstrate a 48-multiplexed sequencing library preparation per reaction tube and prepared 720 bMDA libraries using only 15 reaction tubes. Even with increased multiplexity, the technical performance, such as amplification uniformity, coverage breadth, and false-positive mutation detection rate, remain similar to that of conventional MDA. Importantly, the single-cell bMDA data show sufficient genetic coverage to perform single-nucleotide resolution genome analyses, such as the detection of SNV, SVs, and kataegis. To apply bMDA to integrative spatial genomics, we used the spatially resolved laser-activated cell sorting (SLACS) device[32] and demonstrate its applicability in triple-negative breast cancer (TNBC) to simultaneously analyze CNA, SNV, SV, and kataegis in multiplexes. This has enabled a comprehensive understanding of the spatial genomic landscape of tumors. TNBC tissues were analyzed at a depth of ~5000× with a spatial resolution. Using bMDA to analyze these tissues reduced the cost of library preparation for integrative spatial genomics by $1/N$, thereby demonstrating the scalability and potential of bMDA in spatial genomics (Fig. 1e, f).

## Results

### Dealing with MDA inhibition mediated by barcoded primers

We added barcode sequences at the 5′ end of a conventional MDA primer (random hexamer; N6) (Fig. 1c) to examine whether this would significantly alter the MDA reaction. We further introduced universal PCR handle sequences to selectively enrich the barcoded DNA fragments after MDA. Thus, our trial barcoded primers consisted of a cascade of Illumina Read 1 (15-mer), cell barcodes (8-mer), and random hexamer sequences (6-mer), named R15B8N6 (29-mer in total) (Supplementary Fig. 1a, b). A bMDA experiment with the designed R15B8N6 primer confirmed that primer modification resulted in the inhibition of the MDA reaction (Supplementary Fig. 1c); to the best of our knowledge, an appropriate model to explain this phenomenon has not yet been developed. After testing various hypotheses (Supplementary Note 1), we found that the increased length of the barcoded primer and the unusually high concentration of the primer in the MDA reaction (100× that in PCR) were the primary factors that inhibited the bMDA reaction (Supplementary Fig. 1d, e). The effects of these two factors were further quantitatively evaluated, revealing global trends of decreased MDA amplification efficiency with increasing length or concentration of the barcoded primer (Supplementary Fig. 2a, b). The reduced amplification efficiency was partially mitigated by increasing the enzyme quantity to a certain extent (up to 2×) (Supplementary Fig. 2c). Based on these results, we formulated hypothetical models to understand the mechanism underlying MDA inhibition caused by the abovementioned prime factors and not by random hexamers (Supplementary Note 2 and Supplementary Fig. 2d, e). The models indicated that inhibition originated from the intrinsic properties of the polymerase and primers; therefore, we reduced the concentration and length of the barcoded primer.

However, this significantly affected the functionality of bMDA. Thus, we need to engineer the level of reduction to consider the tradeoffs and simultaneously achieve a bMDA amplification efficiency that is relatively similar to that of conventional MDA. First, a reduction in primer length can diminish the functionality of the primer, including (1) a cell barcode and (2) a motif for subsequent barcoded product enrichment. To substitute the role of Illumina Read 1 (R15) in subsequent PCR-based enrichment of the barcoded product, we introduced a 5′ biotin modification as a capturing motif to substantially reduce the primer length. The length of the cell barcode was also reduced from 8-mer to 6-mer to achieve a bMDA amplification slope of 82% compared to that of MDA (Supplementary Fig. 2b). The final barcoded primer was composed of a 5′ biotin modification, cell barcodes (6-mer), and random hexamers (Supplementary Fig. 2f), and was named bB6N6.

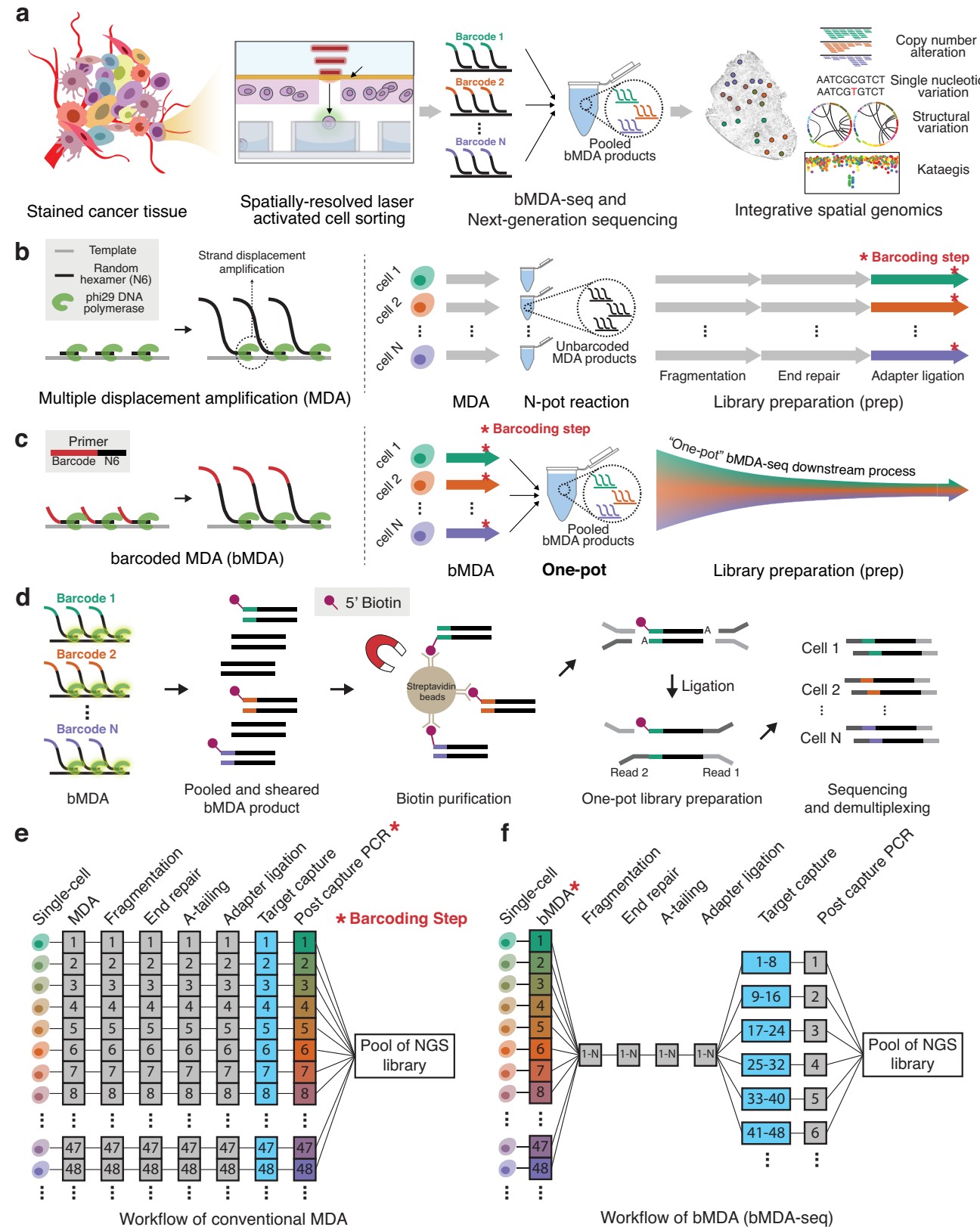

Secondly, a reduced concentration of barcoded primers could decrease the number of barcoded DNA products post-bMDA. If the concentration is reduced significantly, the resulting bMDA library may fail to cover the entire human genome, thereby precluding high-coverage genome analysis. Considering this trade-off, we decided to use 2% of the barcoded primers out of the total N6-containing primers

(1 μM bB6N6 and 49 μM N6) (Supplementary Fig. 2g). Typical MDA amplifies the single-cell genome ~4,000,000-fold; therefore, bMDA was estimated to cover the entire human genome at a depth of ~700×, even though the concentration of the barcoded primer was only 2% (Supplementary Note 3 and Supplementary Fig. 3a–d). Additionally, the amplification efficiency of bMDA was comparable to that of

**Fig. 1 | Barcoded MDA (bMDA) enables multiplexed preparation of a single-cell genome sequencing library, realizing cost-effective and higher-throughput single-nucleotide resolution single-cell genome analysis. a** bMDA-seq can expand our understanding of heterogeneous cell populations by allowing a large number of single cells to be analyzed in multiplex and at single-nucleotide resolution. Post bMDA-seq, each single-cell data showed a genome coverage depth that was sufficient to perform integrative spatial genomics. **b** Conventional multiple displacement amplification (MDA) involves preparing a sequencing library individually for each single-cell because a cell barcode is incorporated at the end of the library preparation step (marked as *). **c** Schematic of bMDA and its workflow bMDA-seq. Since bMDA uses a barcoded primer instead of the conventional random hexamer. The cell barcode used for sample demultiplexing is incorporated during the bMDA reaction (marked as *). After pooling the barcoded MDA products, a sequencing library can be prepared in a single reaction tube (one-pot), thereby reducing the number of library preparation reactions according to the barcoding capacities. **d** Schematic illustration of bMDA-seq workflow. After performing MDA using the barcoded primer, bMDA products were pooled in a single reaction tube. Pooled bMDA products were first fragmented to obtain a desired library insert size. Then, only barcoded DNA fragments were enriched using streptavidin–biotin interactions. Finally, ligation-based library preparation was performed in one-pot to obtain the final NGS sequencing library. **e, f** Detailed procedures of conventional MDA and bMDA show that the multiplexing capability of bMDA remarkably reduces reagent costs and labor required for integrative spatial genomics.

conventional MDA when the proportion of bB6N6 primers was less than 2% (Supplementary Fig. 2b). We maintained the total amount of N6-containing primers (i.e., the sum of the barcoded primer and random hexamer) while reducing the concentration of the barcoded primer because a reduction in the N6 concentration resulted in an increased amplification bias owing to the reduced number of strand displacement events[20,33] (Supplementary Fig. 1d).

The overall workflow, including bMDA and its downstream processes, using the newly designed bB6N6 barcoded primers, is shown in Fig. 1d. We used the term bMDA-seq workflow to distinguish it from bMDA, which solely represents the MDA reaction using a barcoded primer. bMDA-seq was performed following the conventional ligation-based library preparation method, with slight modifications to include the pooling and enrichment of biotinylated (or barcoded) DNA fragments. Notably, the cell barcode is designed to be found only at the beginning of NGS Read 2 (Supplementary Fig. 3e) because 5′ biotin modification blocks the ligation of biotinylated barcoded DNA strand with the Illumina adapter's Read 1 strand; thus, calling of the barcode sequences becomes convenient post sequencing. Moreover, a biotin enrichment method was developed to minimize double-stranded DNA denaturation during the elution of biotinylated DNA fragments (Supplementary Table 1 and Supplementary Note 4).

### Correction of barcode bias and the multiplexing capability of bMDA-seq

To demonstrate the feasibility of bMDA-seq, 48 different barcode sequences of 6 nt length were designed to satisfy the minimum base substitutions between any barcode pair of three nucleotides (Supplementary Table 2, "Methods"). Next, using the designed barcode, we tested whether the presence of a specific barcode sequence at the 5′ end of the primer could cause any systematic bias in the bMDA-seq workflow. We predicted that differences in barcode sequences would cause variations in the number of barcoded DNA fragments after bMDA or library preparation (Fig. 2a and Supplementary Note 5). Using 48 biotinylated barcoded primers, bMDA products were processed after equivolume pooling. We observed that the number of NGS reads belonging to a specific barcode varied among the different barcodes, with a coefficient of variation (CV) of 29.9% (Fig. 2b and Supplementary Fig. 4a).

To resolve this issue, which we termed barcode bias, we hypothesized that the variation is barcode-specific, and the number of barcoded DNA fragments generated is proportional to the concentration of the barcoded primer. The linearity hypothesis was experimentally validated (Supplementary Fig. 4b), and the concentration of the barcoded primer was fine-tuned by changing the ratio between bB6N6 and N6, without modifying the total concentration of the N6-containing primers (Fig. 2a). After three cycles of barcoded primer concentration balancing, a CV of 7.68% was achieved (Fig. 2b, Supplementary Fig. 4a and Supplementary Table 2), which was comparable to the variation observed using the conventional concentration-based Illumina library pooling strategy (CV = 8.53%; $n = 107$ samples). Barcode bias depended largely on the

barcode sequences; therefore, the normalization was reproducible between experimental replicates and was unaffected by alterations in the concentration of the template used for the bMDA reaction (Supplementary Fig. 4c). Thus, we conclude that barcode bias could be successfully corrected by fine-tuning the barcoded primer concentration.

Next, the barcoding status of the bMDA-seq library was examined. Six primers were arbitrarily selected from the 48 bMDA primers, and bMDA-seq was performed for each barcode without pooling. Single-plex bMDA-seq was performed to exclude confounding effects that might arise from interactions between differently barcoded bMDA products. Most NGS reads (82.5%) contained the expected barcode sequence at the beginning of NGS read 2 (Fig. 2c and Supplementary Fig. 4d, e). A detailed examination of the remaining non-barcoded reads is provided in Supplementary Note 6.

To guarantee the absence of meaningful barcode swapping in the bMDA-seq process, we amplified the human genome using 47 barcodes and the mouse genome using the remaining barcodes. After pooling and library preparation, the barcode-swapping ratio was determined by analyzing whether the barcoded NGS reads were mapped to the human or mouse genome. We found that the probability of finding human genome sequences for barcodes used for mouse genome amplification ($0.33 \pm 0.13$% s.e.m.; $n = 6$) was less than or equal to the previously reported ratio of Illumina index hopping ($0.58$%)[34], which is an incorrect assignment of an NGS library index to a different index (Fig. 2d). The probability of finding mouse genome sequences for barcodes used for human genome amplification ($8.9 \times 10^{-3} \pm 7.0 \times 10^{-4}$% s.e.m.; $n = 6$) was much lower than that in the opposite scenario. This can be explained by the fact that the number of barcodes amplifying the human genome was 47 times that of the mouse genome. We also confirmed that the CNA of the bMDA-amplified products matched those of the bulk (Supplementary Fig. 4f, g). From these results, we conclude that bMDA is an adequate multiplexing technology based on three criteria: (1) it shows a uniform read count across barcodes, (2) most reads contain the expected barcode sequences, and (3) the barcode swapping ratio is low.

### Performance of bMDA was comparable to conventional MDA

To demonstrate the applicability of bMDA for single-nucleotide resolution single-cell genome analysis, we isolated single HL-60 cells using a phenotype-based high-throughput laser-aided isolation and sequencing (PHLI-seq) platform[4] (Fig. 3a). PHLI-seq is an infrared (IR) laser-based high-throughput cell isolation technique that does not cause DNA damage and is useful for spatially resolved omics analysis. bMDA for each isolated single-cell and for comparison–in-house MDA, with a protocol similar to that of bMDA, except for the primer (MDA) and MDA using a commercial kit (MDA kit)–were performed concurrently.

bMDA showed high genome coverage breadth ($86.7 \pm 0.97$% s.e.m.; $n = 3$), and the result was comparable to that of in-house MDA ($84.7 \pm 3.3$% s.e.m.; $n = 3$) and MDA kit ($81.1$%; $n = 1$) (Fig. 3b and Supplementary Fig. 5a). This result is better than the low coverage breadth

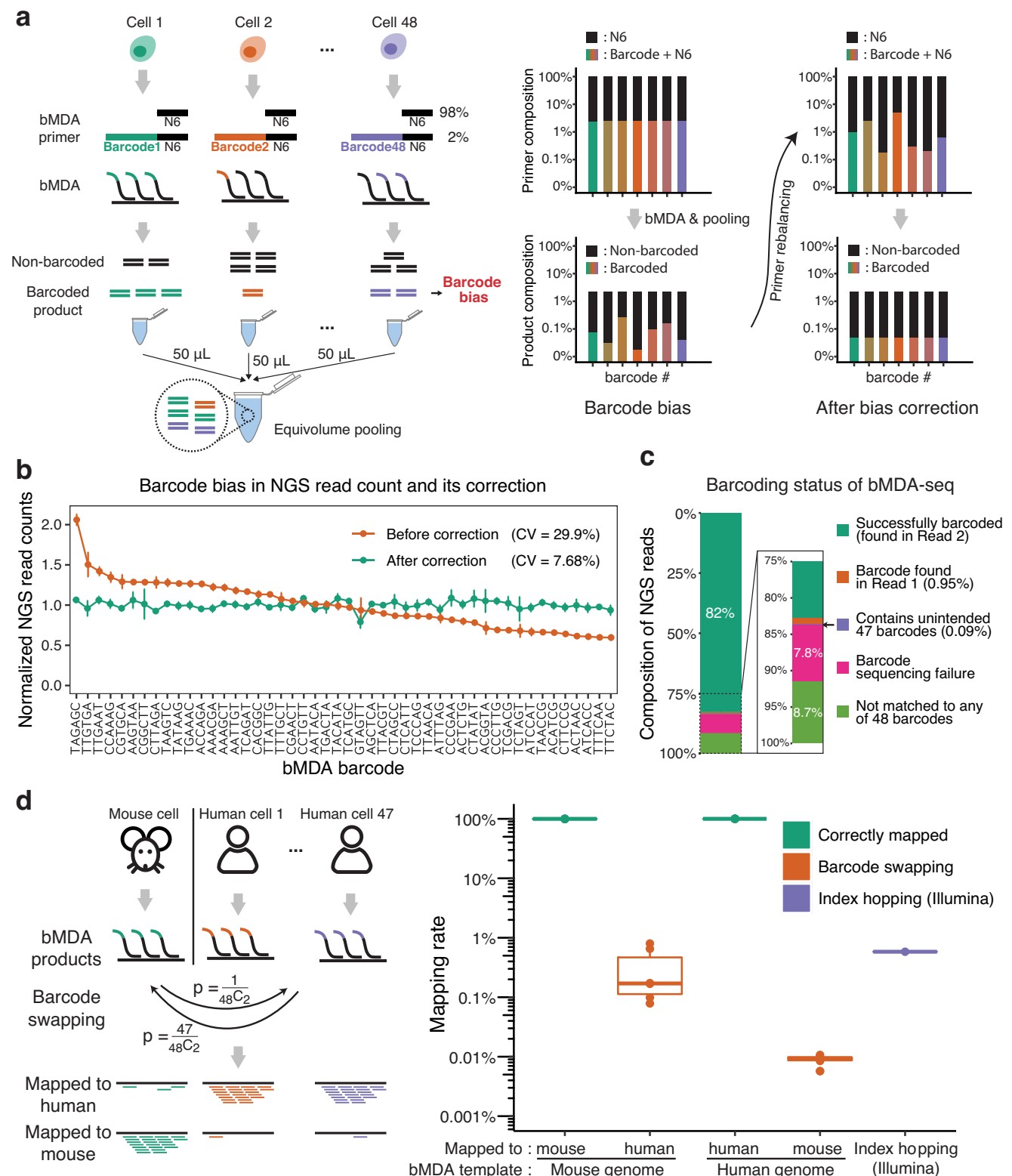

of tagmentation-based sparse single-cell genome analysis methods (Tn5-based)[35–37] (21.6 ± 3.7% s.e.m.; $n = 2$) (Fig. 3b). Although Tn5-based technologies markedly increased the throughput and data quality of single-cell genome analysis by omitting pre-amplification, the trade-off of losing the ability to analyze the single-cell genome at a single-nucleotide resolution was successfully accomplished (Supplementary Fig. 6). We focused on increasing the multiplexity of single-cell genome analysis methods while retaining the capability of single-

nucleotide resolution genome analysis (Fig. 3c and Supplementary Fig. 5b).

Next, the coverage uniformity of bMDA was evaluated using a Lorenz curve (Fig. 3d) to assess the uniformity of genome amplification by bMDA. The area under the Lorenz curve (AUC) was used to evaluate coverage uniformity, and we observed no significant difference between bMDA and conventional MDA in the 66 pg ($p = 0.1$, Wilcoxon rank-sum test) and single-cell groups ($p = 0.14$, Student's $t$ test)

**Fig. 2 | bMDA-seq was optimized and validated with cell line gDNA. a** Schematic illustration of the equivolume pooling approach, barcode bias, and its correction. **b** NGS read count before and after barcode bias correction shows that the barcode bias can be successfully corrected by fine-tuning the proportion of barcoded primers. The NGS read count of a specific barcode was normalized to -1 to obtain the normalized NGS read counts (*y*-axis). Points and lines represent the mean ± s.e.m. (*n* = 3 independent experiments). **c** Barcoding status of the bMDA-seq library was analyzed using NGS, showing that the majority of NGS reads (82.5%) were barcoded as predicted. Other unpredicted NGS reads were classified by their barcode sequences, and are shown as a bar chart. To eliminate the putative confounding effect that may arise after the pooling of differently barcoded bMDA products, a single bMDA product was used for the analysis. The ratios from each bMDA product (*n* = 8) were averaged to obtain a representative single plot. **d** The barcode

swapping ratio of bMDA-seq was sufficiently lower than the other well-known index swapping ratios, such as index hopping in multiplexed Illumina sequencing. To assess the barcode swapping ratio, 47 out of 48 barcodes were used for amplifying the human genome (HL-60), and the remaining barcode was used for amplifying the mouse genome (NIH3T3). Notably, of the $_{48}C_2$ possible barcode swapping cases, the probability of swapping from human to mouse is 47 times higher than the opposite due to its higher number of possible swapping cases. Number 1 vs. number 47 was chosen to simulate the worst-case scenario. Box plot show the median (center line), first and third quartiles (box edges), while the whiskers extend from the box edge to the largest or smallest value no further than 1.5 times the interquartile range (IQR) from the box edge (*n* = 6 independent experiments). Source data are provided as a Source Data file.

(Fig. 3e). bMDA showed a slight tendency to decrease the AUC, which could be attributed to the effect of the barcoded primer on the MDA reaction. However, this decrease was smaller than the intragroup variation (cell-to-cell variation) in the bMDA, MDA, and MDA-kit groups (one-way ANOVA, *p* = 0.15).

To evaluate the accuracy of the CNA calls, the detected CNA was displayed by decreasing the AUC order. No notable differences were observed between bMDA and conventional MDA (Fig. 3f). CNAs were successfully called for all samples amplified from 66 pg (10-cell equivalent) of gDNA. However, only a few high AUCs showing single-cell data exhibited reliable CNA calls (Supplementary Fig. 5c), and most single-cell MDA products exhibited high false-positive CNA detections compared to bulk CNA profiles. Single-cell bMDA, MDA, and MDA kit groups showed high false-positive CNA calls; thus, the false detections were mainly due to a fundamental limitation of MDA-based chemistry[38,39], rather than the modified primer in bMDA.

Finally, the allelic dropout (ADO) rate and false-positive mutation detection rates (FPR) were assessed to determine whether bMDA can accurately detect SNV. ADO rate of bMDA (13.2 ± 1.6% s.e.m.; *n* = 3) was similar to that of in-house MDA (19.0 ± 5.1% s.e.m.; *n* = 3) and the values of previous studies (7–43%) (Fig. 3g)[4,30]. The FPR of bMDA ($1.1 \times 10^{-5}$) was also comparable to that of in-house MDA ($0.99 \times 10^{-5}$) and previously reported methods ($2 \times 10^{-5}$–$3 \times 10^{-5}$)[4,30]. Furthermore, the variant allele frequency (VAF) of the detected heterozygous single-nucleotide polymorphisms (SNPs) confirmed that the performance of bMDA and in-house MDA was highly similar in terms of evenly amplifying the two different alleles in a single-cell (Supplementary Fig. 5d).

## Application of bMDA in TNBC revealed a single-nucleotide resolution mutational landscape

We applied bMDA-seq to TNBC tissues to demonstrate its potential in resolving tumor heterogeneity at a single-nucleotide resolution. TNBC tissue sections were acquired from two patients with TNBC (T1 and T2). Using PHLI-seq[4], we isolated 20 and 28 cell clusters from T1 and T2 patients, respectively. Each cell cluster consisted of ~20 cells, and with guidance from H&E-stained tissue images and pathologist expertise, we achieved an average tumor purity of 98.8% (Supplementary Figs. 7 and 8a, b). The 48 isolated cell clusters were processed simultaneously using a 48-plex bMDA-seq workflow.

By isolating spatially adjacent cell clusters instead of individual single cells, we mitigated the inherent amplification bias and errors associated with MDA[40] (Fig. 4a–d, Supplementary Fig. 8c–e and Supplementary Table 3). This approach yielded a significantly improved sensitivity of 91.0 ± 0.64% (s.e.m.; *n* = 27) for SNV detection, surpassing the sensitivities of 77.8 ± 3.5% (s.e.m.; *n* = 10) of single-cell MDA and 8.6 ± 2.0% (s.e.m.; *n* = 6) of single-cell Tn5 amplification (Fig. 4b). Moreover, we achieved a high sensitivity of 95.5 ± 2.5% (s.e.m.; *n* = 10) and a specificity of 98.8 ± 0.8% (s.e.m.; *n* = 10) for CNA detection (Fig. 4c and Supplementary Fig. 8e). The quality control (QC) pass ratio, which measures amplification uniformity and CNA detection accuracy, reached 92.6%, in contrast to the pass rate of less than 10%

observed in single-cell analyses (Supplementary Fig. 8f, g). Consequently, our approach enabled concurrent detection of CNAs, SNVs, and SVs in each cell cluster with a high degree of confidence.

For orthogonal validation, we first compared the bMDA-seq CNA data with data generated by MDA in patients with the same cancer. Hierarchical clustering of the bMDA-seq and MDA data showed that the data from each method were intermixed, suggesting high agreement between the two methods (Supplementary Fig. 9a).

Next, we integratively analyzed the genomic landscape of the T1 tumors in terms of CNA, SNV SV, and kataegis (Fig. 4e–i). Interestingly, the hierarchical clustering results of microniches differed according to the CNA, SNV, SV, and kataegis data. While the SNVs revealed the presence of three major subclones, additional analysis of CNA, SV and kataegis further delineated each subclone (Fig. 4f–i). This result indicates that different types of genomic variations can contribute to the identification of finer subclonal landscape. Specifically, the discrepancy between CNAs and SNVs in inferring evolutionary relationships underscores the importance of utilizing an accurate method such as bMDA to study spatial genomics. Thus, to determine the evolutionary relationships between microniches in spatial genomics, an integrative approach must be incorporated to gain a comprehensive understanding of spatial genomic landscapes.

The same integrative spatial genomic analysis of T2 tumors provided another extensive view of the evolutionary landscape of the tumor (Fig. 5a). CNA (Fig. 5b), SNV (Fig. 5c, d), and SV analyses (Fig. 5e) revealed four major subclones within the same tumor, and the spatial location of the microniches showed location-specific development of these distinct subclones (Supplementary Fig. 9b). Spatial auto-correlation analysis using first two principal components of the SNV mutational profile yielded Moran's I statistics of 0.484 and 0.478, indicating a significant clonal expansion of tumor cells in adjacent spatial regions. We also analyzed the spatial relationship between CNA and SNV allele frequencies (Supplementary Fig. 9c). In some genes, including *NF1*, a lower copy number correlated with higher SNV allele frequencies. However, genes, such as *LRP1B* showed an increase in SNV allele frequency with increasing copy number. The latter seems to be more common because an increase in gene copy number seems to correlate with an increase in SNV allele frequencies, simply because there are more mutated copy numbers. However, the phenomena in the case of *NF1* provide insight that there may be other cases where the SNV may affect the low copy number, or vice versa.

Interestingly, there was a consistent kataegis on chromosome 12 *ERC1* gene in the upper microniches of the tumor (Fig. 6a and Supplementary Fig. 10). When analyzing the SV of the regions with kataegis within these populations, we observed gene translocations between *ERC1* and *TCOF1* (Fig. 6b, c). In addition, copy number amplification was observed in the same region (Fig. 6d). However, populations lacking kataegis on chromosome 12 did not show copy number amplification or gene translocation (Fig. 6e). We observed other notable SV on chromosomes 12 (Fig. 6f) and 5 (Fig. 6g) in terms of duplications, deletions, inversions, and translocations. Figure 6h, i

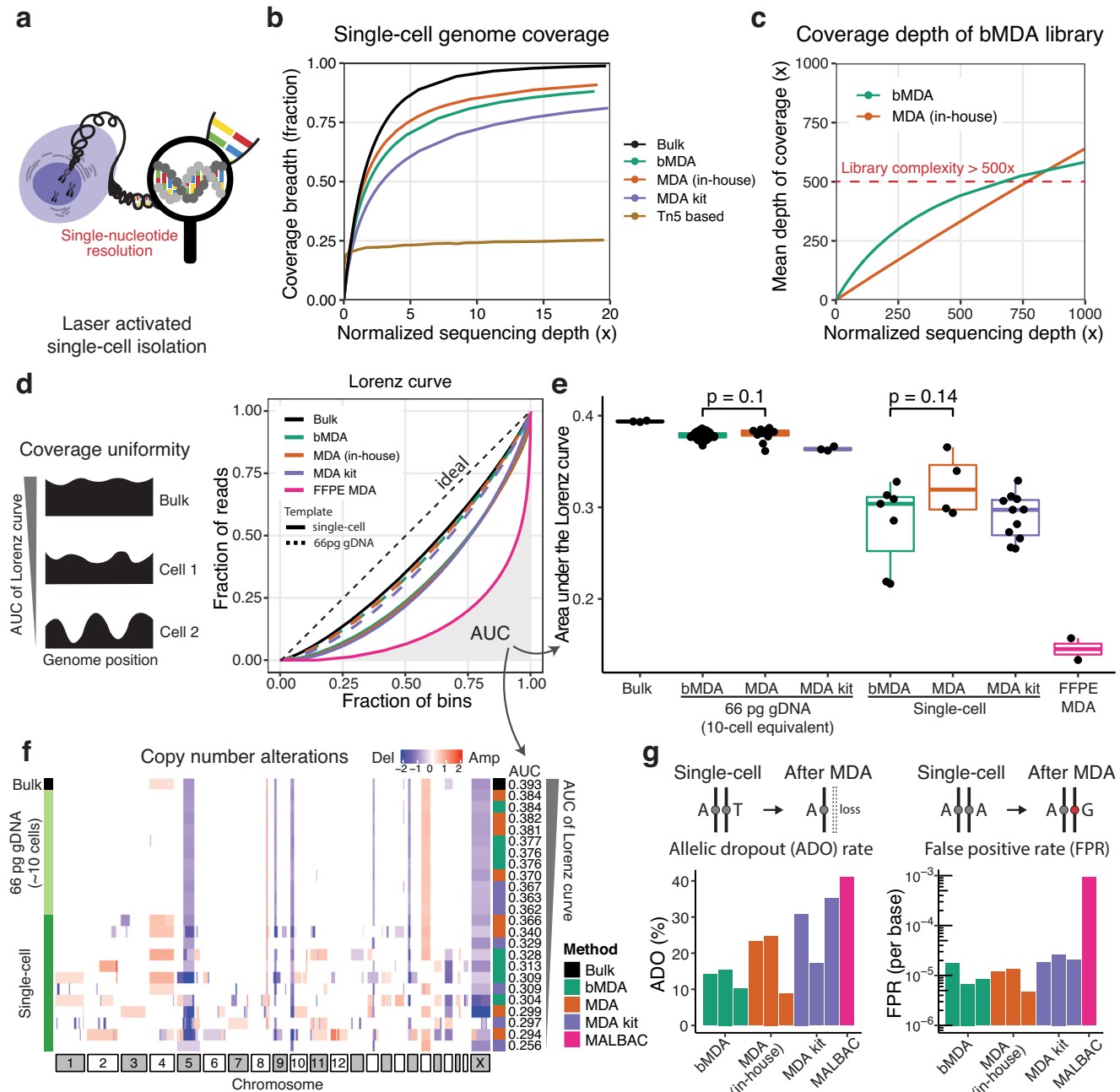

**Fig. 3 | Technical performance of single-cell bMDA is comparable to that of conventional MDA. a** Single cells were isolated by the PHLI-seq platform[4]. **b** Single-cell bMDA showed nearly full genome coverage breadth (0.87×) in contrast to the Tn5-based method (0.22×). bMDA also showed similar coverage breadth compared to the in-house MDA, whose protocol was the same as bMDA except for using N6 as a primer (MDA), or MDA using a commercialized kit (MDA kit). **c** Targeted deep sequencing confirmed that the library complexity of bMDA is high enough for single-nucleotide resolution genome analysis, covering the single-cell whole genome at a depth of more than 500×. **d, e** Amplification uniformity of bMDA and conventional MDA was similar in both the 66 pg ($p = 0.1$ and 95% confidence interval (CI) −0.005, 0.0006, two-sided Wilcoxon rank-sum test) and single-cell groups ($p = 0.14$, $t = -1.6$, degree of freedom (df) = 9, 95% CI −0.10, 0.02, and Cohen's $d = 1.00$, unpaired two-sided Student's $t$ test). Box plot show the median (center line), first and third quartiles (box edges), while the whiskers extend from

the box edge to the largest or smallest value no further than 1.5 times the inter-quartile range (IQR) from the box edge. $n = 3, 98, 11, 3, 7, 4, 11$, and 2 biologically independent samples from the left of the box plot. **f** Copy number alterations (CNA) plot demonstrates that there is no notable difference between bMDA and other conventional MDA methods in the aspect of resolving CNAs. Each row indicates individual MDA-amplified products obtained by different methods and templates. The values displayed on the heatmap represent the log2-transformed relative changes in copy number compared to the average copy number across the entire genome. **g** To confirm the single nucleotide variant (SNV) detection performance, the allelic dropout (ADO) rate and false-positive mutation detection rate (FPR) were evaluated. The bMDA, MDA, and MDA kit results showed comparable performance. bMDA barcoded MDA, MDA in-house MDA with random hexamer, MDA kit commercial MDA kit. Source data are provided as a Source Data file.

summarizes the visual comparison of chromosomal aberrations between the two groups of spatial microniches.

Furthermore, by integrating all chromosomal aberrations (Supplementary Data 1), we constructed a spatial map depicting the

inferred evolutionary relationships between these microniches (Fig. 7). While subclone c1 seems to serve as the ancestral lineage and sequentially gives rise to subclones c2, c3, c4, and, ultimately, c5, it is important to emphasize that the tumor section represents a snapshot

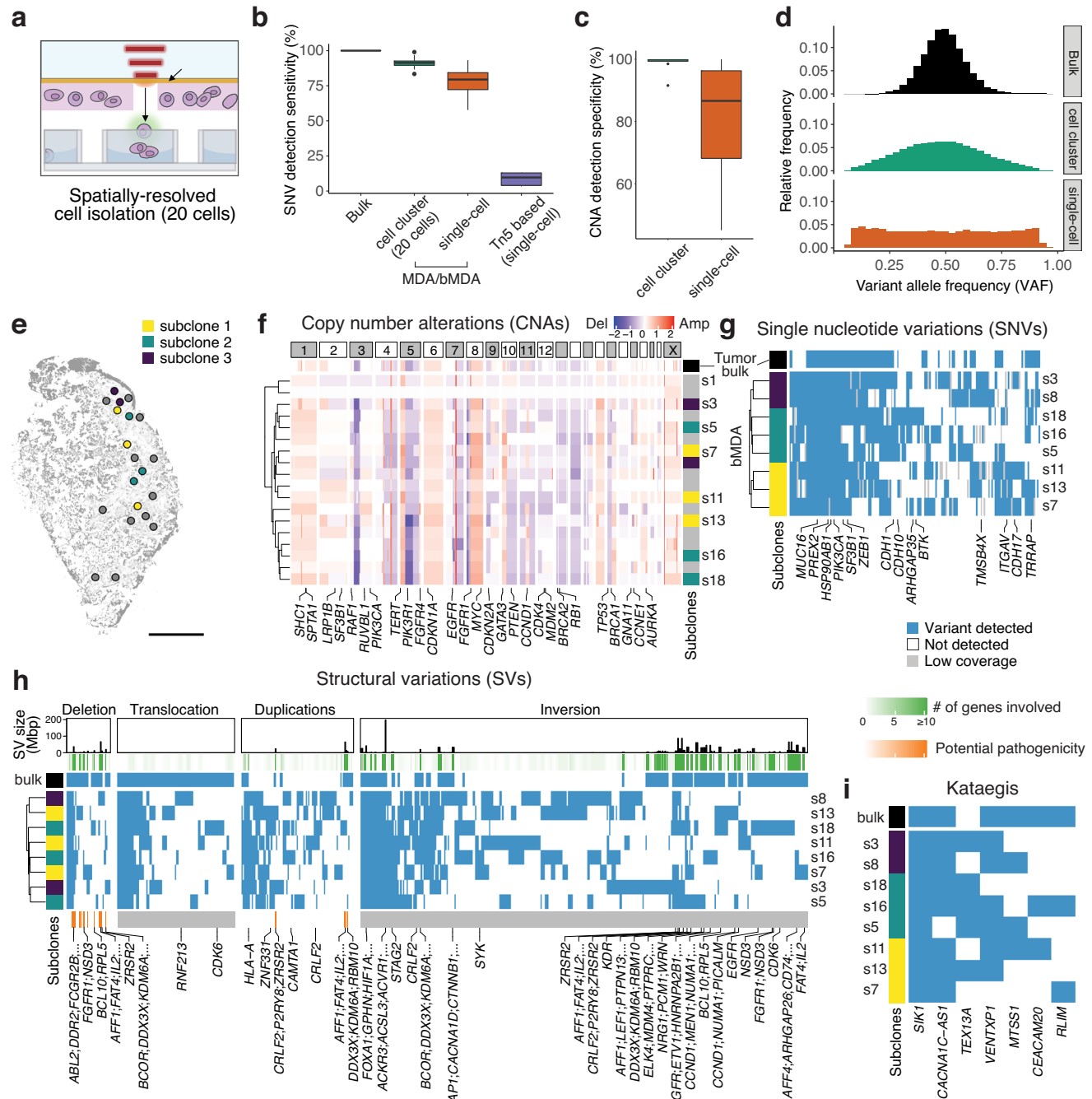

**Fig. 4 | Structural variation and kataegis further delineate clonal evolution of T1 triple-negative breast cancer tissue section than when the subclonality is determined by copy number alterations and single nucleotide variations.**
**a** Isolating spatially adjacent cell clusters provides highly confident integrated genomics. **b** SNV detection sensitivity was higher and less biased for cell clusters compared to single-cell. The SNV sensitivity of cell clusters was calculated by comparing the germline mutations identified in the bMDA-seq data from T1 and T2 tumors to the germline mutations detected in the normal bulk sequencing data (*n* = 27 biologically independent samples). The value for single-cell was obtained by performing MDA/bMDA (*n* = 10 biologically independent samples) and Tn5-based methods (*n* = 6 biologically independent samples) on a cell line. **c** CNA detection specificity was higher and less biased for cell clusters (*n* = 10 biologically independent samples) compared to the single cells (*n* = 26 biologically independent samples). All box plots show the median (center line), first and third quartiles (box

edges), while the whiskers extend from the box edge to the largest or smallest value no further than 1.5 times the interquartile range (IQR) from the box edge. **d** Variant allele frequency distribution of detected heterozygous SNPs was less biased and resembled more to bulk distribution in cell cluster isolation. **e** Spatial landscape of different microniches of a triple-negative breast cancer tissue section. Subclone information was inferred by analyzing SNVs. The gray color indicates that spatial microniches were not included in the whole exome sequencing analysis (scale bar = 1 mm). **f** Heatmap illustrating copy number alterations in different micro-niches. The values displayed on the heatmap represent the log2-transformed relative changes in copy number compared to the average copy number across the entire genome. **g** Single nucleotide variations of different subclones. **h** Structural variations of the different subclones within the same tumor. **i** Genes where kataegis was detected are displayed. Source data are provided as a Source Data file.

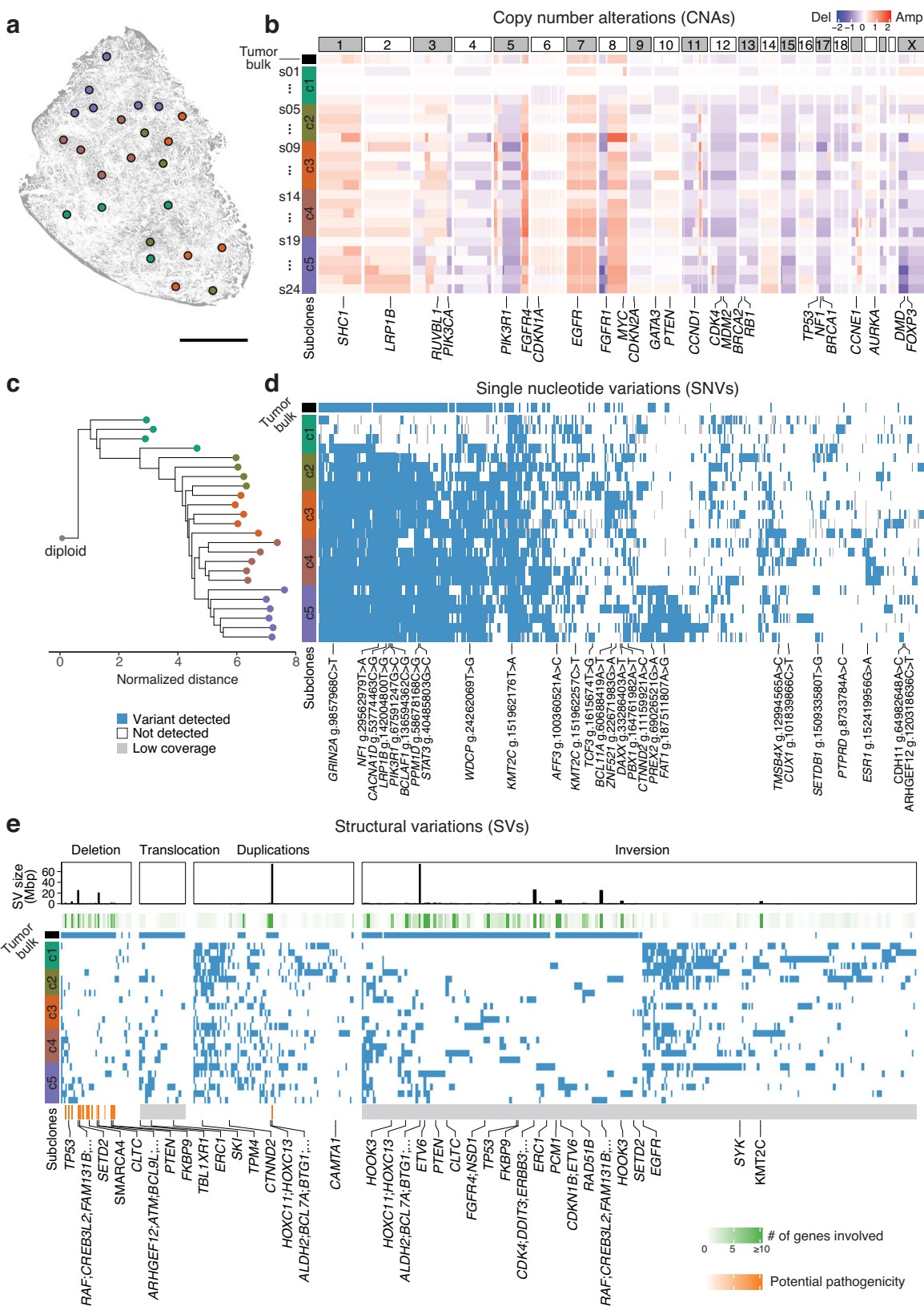

**Fig. 5 | Integrative spatial genomics in T2 breast cancer.** The color and order of each spatial microniche remain consistent throughout the figure. **a** Spatial landscape of different subclones of a triple-negative breast cancer tissue section. Subclone information was inferred through phylogenetic analysis of SNVs (scale bar = 1 mm). **b** Copy number alterations of different subclones. The values displayed on the heatmap represent the log2-transformed relative changes in copy number compared to the average copy number across the entire genome. **c** The phylogenetic tree illustrates the evolutionary relationships among different microniches within the same tumor microenvironment. **d** Single nucleotide variations of different microniches. **e** Structural variations of the different microniches within the same tumor. Source data are provided as a Source Data file.

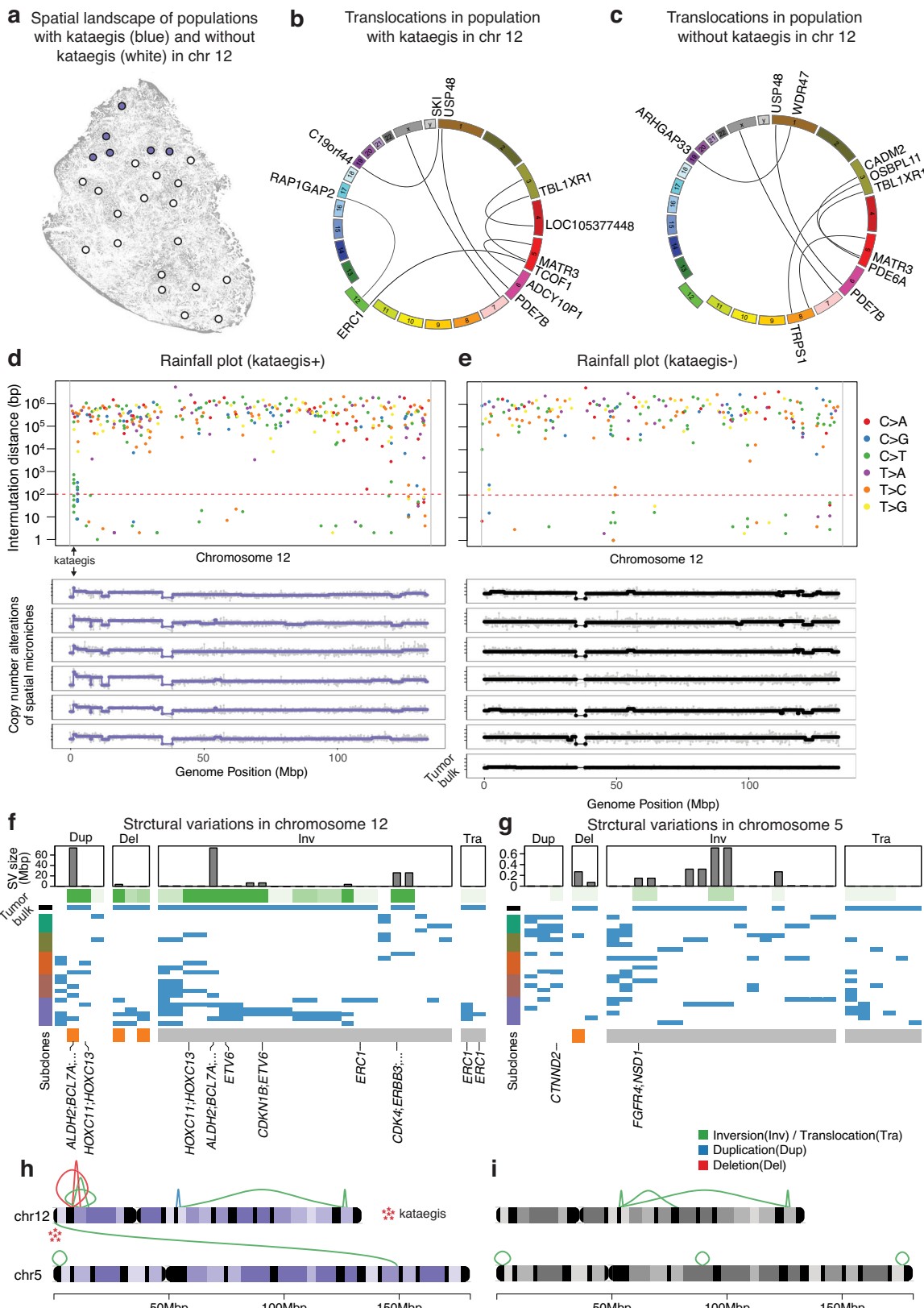

**Fig. 6 | Integrative spatial genomics of T2 tumor reveals the relationship between kataegis, copy number, and translocation in chromosome 12, thereby providing comprehensive insight into the spatial subclone in cancer. a** Spatial landscape of populations with and without kataegis in chromosome 12. **b**, **c** Translocations in two different populations are displayed. **d**, **e** Rainfall plots of the two different populations and copy number alterations in chromosome 12. The region where kataegis occurred had copy number amplification. **f**, **g** Structural variations in Chromosome 12 and 5. **h**, **i** Schematic display of genomic aberrations in chromosomes 12 and 5. Source data are provided as a Source Data file.

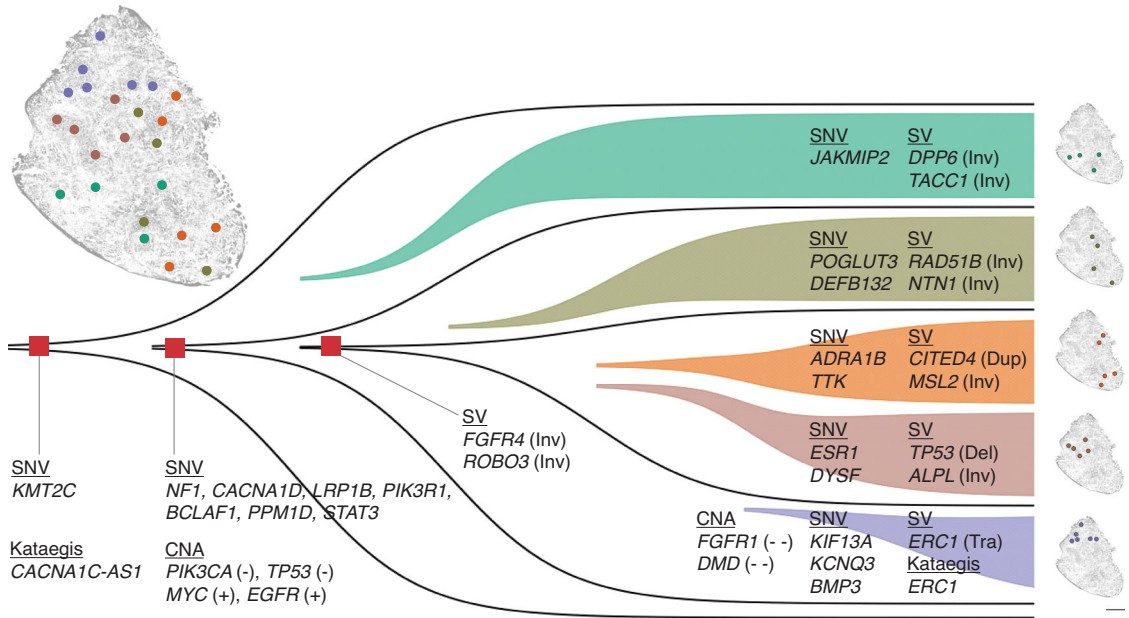

**Fig. 7 | bMDA facilitates integrative spatial genomics, suggesting a more plausible tumor evolutionary history of T2 triple-negative breast cancer.** Source data are provided as a Source Data file.

of the tumor's evolution. These subclones are believed to have undergone diversification over time (Fig. 7), leading to their observed spatial distribution. Rather than positing a specific evolutionary direction, this analysis provides valuable insights into the spatial relationships and potential evolutionary dynamics of tumor subclones within tissues.

In addition, in the two analyzed tumors, we revealed significant genetic nuances that were overlooked by bulk sequencing. First, although not classified as actionable, variations in the *ESR1* gene often guide the use of elacestrants. Notably, we identified some subclones in the T2 tumor exhibiting an *ESR1* p.R548H mutation that were missed in bulk sequencing (Supplementary Data 1). Intriguingly, we uncovered heterogeneous *CCNE1* amplification patterns associated with therapies such as RP-6306 and BLU-222. In T1 tumors, bMDA-seq revealed subclones with *CCNE1* amplification that were absent from the bulk sequencing data (Fig. 4f). In T2, both *CCNE1* amplification and deletion were observed in the subclones, despite the bulk data showing amplification (Fig. 5b). This finding illustrates the importance of acknowledging intra-tumoral heterogeneity when considering therapeutic strategies. As precision oncology advances with more mutation-matched treatments, technologies, such as bMDA-seq, will be instrumental in guiding personalized treatment strategies by revealing the intricate subclonal architecture of tumors. Detailed interpretations of the detected mutations are summarized in Supplementary Note 7.

## Discussion

In this study, we developed a technology that effectively increased the multiplexity of integrative spatial genome analysis. Multiplexing chemistry was realized by incorporating barcodes during the bMDA reaction, and the major obstacle to MDA reaction inhibition was resolved by reducing the length and concentration of the barcoded primers. Consequently, the cost of library preparation, which is a major bottleneck in large-scale single-cell genomics, was reduced to an almost negligible level. Despite these improvements, the technical performance of integrative spatial genome analysis remains similar to that of conventional MDA. Because the protocol does not require specialized equipment, the approach is simple to adopt by general laboratories and robust enough to be widely used while allowing for

the analysis of hundreds to thousands of single cells. The potential of this spatial genomic analysis using bMDA is that researchers can not only infer evolutionary relationships between different intra-tumoral subclones but also search for characteristic features with different types of genomic aberrations occurring simultaneously. The demonstration at T2 showed that potential chimeric protein sequences could be identified when the chromosomal structure and single-nucleotide sequences were considered simultaneously. bMDA is a scalable, robust, and highly sensitive method for revealing such aberrations simultaneously and will provide a useful tool to dissect the spatial genomic landscape of different biological systems.

Tn5-based technologies[35–37] have been recently introduced to increase the multiplexity of single-cell genome sequencing. Tn5-based methods allow only sparse interrogation of the single-cell genome, whereas bMDA allows single-nucleotide resolution single-cell genome analysis. In addition, when comparing bMDA with other promising single-cell technologies such as Tapestri™ (Mission Bio)[17] and linear amplification via transposon insertion (LIANTI)[41], it should be noted that each approach has its own strengths and limitations. While LIANTI faces challenges in achieving multiplexing chemistry, Tapestri is not ideally suited for exome-wide interrogation of the single-cell genome due to its typical limitation of a maximum target genome region size of 100 kb. Therefore, bMDA has the advantage of simultaneously analyzing a large number of single-cell genomes at single- and exome-wide resolutions. More recently, a method called primary template-directed amplification (PTA)[42] was introduced as an integrated approach that combines amplicon displacement and quasi-linear amplification for whole-genome amplification. PTA has some advantages over MDA, including uniform amplification and high variant calling precision. However, further investigation is required to determine the feasibility of multiplexing their reaction chemistry to simultaneously analyze a large number of tumor microniches. It should also be noted that the use of PTA with cells in tissue sections has not yet been demonstrated and its suitability for spatial genomic applications and barcoding strategies remains to be investigated.

We believe that cancer cells possessing different SNV, SVs, or kataegis, but similar CNA, are important for further decomposing the genomic landscape of tumors. Identifying driver or passenger mutations that distinguish different subclones within the same tumor can

provide significant insights into how tumor cells evolve to form TNBC and guide diagnostic or therapeutic strategies related to these mutations. This analysis can be expanded to simultaneously analyze the spatial subclonal landscape of both primary and metastatic tumors as well as circulating tumor cells[43]. Additionally, cost-effective subclonal profiling facilitated by bMDA enables the construction of personalized circulating tumor DNA panels that target each subclone. Furthermore, SV and kataegis analyses hold promise for revealing the potential benefits of DNA damage repair pathway-targeted therapeutics. Therefore, bMDA-seq-enabled integrative genomic profiling can provide a cost-effective approach for cancer treatment.

The applications of this study can be expanded beyond cancer research. The sequencing of unculturable microbiomes[44], investigating somatic mutations in humans[25,45], and preimplantation genetic diagnosis[46] are only a few examples. Furthermore, by integrating bMDA with single-cell multi-omics technologies, multiplexed profiling of single-nucleotide-resolution single-cell genomes with transcriptomic or epigenomic features can be achieved.

## Methods

### Ethical statement
Our research complies with all relevant ethical regulations. Fresh frozen TNBC tissue sections were obtained from the archives of the biorepository of the Lab of Breast Cancer Biology at the Cancer Research Institute, Seoul National University, Seoul, South Korea. The preparation of tissue was approved by the Institutional Review Board (IRB) at Seoul National University Hospital (No. 1910-130-1072). The patients provided written consent, without compensation, for the archiving and use of tissue and blood samples for research purposes.

### Cell culture and preparation
Human HL-60 cells (CCL-240™) were obtained from ATCC; and human SK-BR-3 cells (cat. no. 30030) and mouse NIH3T3 cells (cat. no. 21658) were obtained from the Korean Cell Line Bank (KCLB) and cultured according to the manufacturer's instructions. HL-60 and SK-BR-3 were cultured in IMDM (Thermo Fisher) with 1% penicillin-streptomycin (Corning) and 10% fetal bovine serum (HyClone) at 37 °C under 95% atmospheric air and 5% $CO_2$. NIH3T3 cells were cultured in Dulbecco's modified eagle medium (DMEM) (Thermo Fisher) supplemented with 1% penicillin-streptomycin and 10% bovine calf serum (HyClone) at 37 °C under 95% atmospheric air and 5% $CO_2$. Adherent cells, such as SK-BR-3 and NIH3T3 cells, were grown to a confluence of 50–80% and subsequently treated with TrypLE (Invitrogen) for five min in order to detach the adherent cells. Next, TrypLE was quenched with an equal volume of the growth medium, and spun down at 415 g for 3 min after transferring into another conical tube. Suspension cells, such as HL-60 cells, were grown to a concentration of $1 \times 10^6$ cells/ml and spun down at 415 g for 3 min. Then, the supernatant of both cell types was removed, and the cells were resuspended in 1 ml of 1× PBS. Spin-down and supernatant removal were performed again to remove any residual culture medium. Cell pellets were resuspended in 10 µl of 1× PBS and spread on indium tin oxide (ITO)-coated glass (Fine Chemicals Industry, Republic of Korea), followed by air drying and 30 s of methanol fixation for single-cell isolation using PHLI-seq. A DNeasy Blood & Tissue Kit (69504, QIAGEN) was used to extract gDNA from the cultured cell line.

### Tissue preparation
Fresh frozen TNBC tissue sections were cryosectioned and thawed on ITO-coated glass slides. For hematoxylin and eosin (H&E) staining, the tissue sections underwent the following procedures: (1) air drying at room temperature for 15 min, (2) rinsing the slide in tap water for 5 min, (3) staining with Harris' hematoxylin solution (Merck) for 3 min, (4) rinsing in tap water via quick dipping, (5) rinsing in 1% hydrochloric acid (HCl) in ethanol solution via quick dip, (6) rinsing in tap water for 5 min, and (7) staining with Eosin Y (BBC Biochemical) for 3 s. Afterward, the stained slides were washed and dehydrated using a series of ten dips in each of the following solutions: (1) water, (2) 70% ethanol, (3) 90% ethanol, and (4) 100% ethanol. A whole-slide image of the prepared slide was obtained using automated microscopy (Nikon Inverted Microscope ECLIPSE Ti-E).

For cell cluster isolation, H&E-stained tissue sections from two patients with TNBC (T1 and T2) were pathologically inspected to distinguish the spatial locations of cancer cells from normal cells. Based on the identified tumor cell locations, tumor cell clusters, each containing an average of 20 cells, were isolated from unstained fresh frozen tissue sections that were serial to the corresponding H&E sections.

### Participant information
T1 tissue was obtained from surgically resected triple-negative inflammatory breast cancer of a 49-year-old woman who had received neoadjuvant chemotherapy (five of six planned cycles of docetaxel and adriamycin due to disease progression). The initial clinical stage before neoadjuvant chemotherapy was cT4N2M0, and the pathological staging post total mastectomy with axillary lymph node dissection in May 2019 was pT2N1M0. There was no evidence of recurrence.

T2 tissue was obtained from surgically resected triple-negative invasive ductal carcinoma of a 50-year-old woman who had received neoadjuvant chemotherapy (four cycles of adriamycin + cyclophosphamide and two of four planned cycles of docetaxel owing to adverse effects and limited response to chemotherapy). The initial clinical stage before neoadjuvant chemotherapy was cT4N3M0, and the pathological staging post total mastectomy with axillary lymph node dissection in April 2019 was pT2N3M0. In July 2019, the patient developed metastases to the brain and liver.

The application of the bMDA method described in this study was demonstrated using data from two female participants. The study design did not specifically consider sex and/or gender as a factor. However, bMDA technology is designed for genomic data analysis and its applicability is not limited by the gender or sex of the study participants. While the study did not explore potential sex-specific effects or differences, the bMDA method can be applied to genomic data from individuals of any sex or gender.

### Target selection in tissue samples
The spatial targets for isolating cell clusters in TNBC tissue section were selected by expert pathologists who aimed to capture as many representative heterogeneous subclones as possible. Their selection was based on morphological characteristics, H&E staining, and other pathological features that suggested the presence of distinct tumor subclones. By selecting representative spatial samples, we ensured that our analysis captured the diverse genomic landscape of the tumor and provided insights into the spatial organization of its subclones.

To predominantly capture tumor cells and limit healthy cell contamination, we carefully selected our target regions. By visually examining H&E stained images, we avoided areas with substantial infiltration of healthy cells. Our goal was to enrich our cell clusters with tumor cells, which would reduce the potential confounding impacts from healthy cell contamination in our later analyses.

### Cell isolation using PHLI-seq
Single HL-60 cells or cell clusters of TNBC tissue sections were isolated as described in the PHLI-seq paper[4]. Briefly, cells were prepared on 100-nm thick ITO-coated glass slides. When an infrared (IR) laser is applied, the ITO layer absorbs the energy of the laser and vaporizes, thereby converting optical energy into physical energy. Owing to evaporation pressure, cells in the region of interest (ROI) were released from the glass slide and transferred to the retrieval cap

strips. For cell retrieval, optical flat eight-strip PCR tube caps (TCS0803, Bio-Rad) were used. An appropriate cell lysis solution was pre-loaded onto the tube cap to transfer laser-isolated cells directly into the solution. By observing the brightfield cell image, we adjusted the region of cells to be isolated in real time or automatically using pre-determined scripts.

### In-house multiple-displacement amplification (MDA) and bMDA

For cell lysis and lysed gDNA denaturation, 1 µl of template containing gDNA or cells was mixed with 3 µl of cell lysis solution (400 mM KOH, 10 mM EDTA, 100 mM DTT), 2 µl of PBS (REPLI-g Single-Cell Kit, Qiagen), and 1 µl of 500 µM random hexamer. Cell lysis and denaturation were performed on ice for 20 min. Subsequently, 3 µl of neutralization buffer, consisting of 400 mM HCl and 600 mM Tris-HCl (pH 7.5), was added to neutralize the lysis buffer. Finally, 40 µl of MDA master mix containing 23 µl of water, 5 µl of 10× phi29 DNA polymerase reaction buffer [500 mM Tris-HCl, 100 mM MgCl$_2$, 100 mM (NH$_4$)$_2$SO$_4$, 40 mM DTT, pH 7.5], 4 µl of 25 mM dNTP, 2 µl of 1 mM random hexamer, 2 µl of phi29 DNA polymerase (Genomiphi V2 DNA amplification kit, Cytiva, cat. no. 25-6600-31), 3.2 µl of 40% (w/v) PEG 8000, 0.25 µl of 1 M DTT, 0.5 µl of 50 µM SYTO™ 13 Green Fluorescent Nucleic Acid Stain (Invitrogen), and 0.05 µl of 500 nM ROX was added. A total of 50 µl MDA reaction mix was incubated at 30 °C for 12 h, followed by inactivation at 65 °C for 10 min. SYTO 13 and ROX fluorescence dyes were added for the real-time monitoring of MDA amplification. The first 3 h of the MDA reaction was monitored in real-time (RT-MDA), and the Applied Biosystems 7500 Fast Real-Time PCR System was used for quantitative monitoring. Samples with an amplification start time of less than 30 min were considered to pass the quality control (QC) and were expected to exhibit low amplification bias, enabling accurate calling of copy number alterations (CNAs). Random hexamers were replaced with the appropriate barcoded primer mix for barcoded multiple-displacement amplification (bMDA).

### Barcoded primer design

To evaluate how various lengths of the barcoded primer influenced bMDA amplification efficiency, shorter barcoded primers were designed by trimming the 5′ end of the longer barcoded primer (R15B8N6).

Barcrawl (v100310)[47] was used to design the 5′ barcode sequences of the bB6N6 primers. Among the 4096 (4⁶) possible barcodes of length 6, barcodes with at least three base differences between the arbitrary pair of barcodes were selected to be tolerant of the sequencing error at the barcode position. Minorly, barcodes that contained homopolymers of length ≥4 or barcodes with GC content greater than 90% or less than 10% were excluded from the candidate barcodes. Thus, the command-line argument for the barcode design was -l 6 -p 4 -g 90 -c 10. Among the 98 designed barcodes, 48 were arbitrarily chosen for bMDA demonstration (Supplementary Table 2).

With the designed barcode sequences, bMDA barcoded primers of sequence /5Biosg/JJJJJJNN NN*N*N were ordered from Integrated DNA Technologies (IDT) with standard desalting purifications. Here, /5Biosg/ represents 5′ biotin modification, and six consecutive Js represent one of the designed barcode sequences.

### Purification and fragmentation of the bMDA products

The bMDA products of equal volume were first pooled, and then purified using 0.8× volume of solid-phase reversible immobilization (SPRI) beads (Celemag Clean-up Bead, Celemics, Republic of Korea) for 1× volume of bMDA products. DNA binding and bead wash were performed according to the manufacturer's instructions, and the bound DNA was eluted with 0.4× volume of water. Since, the viscosity of the eluate increases due to the increased DNA concentration (over 1.5 µg/µl), magnetic separation of the SPRI beads becomes very slow. The solution was centrifuged at 20,000 RCF for 1.5 h at room

temperature to separate the SPRI beads and the remaining eluate was carefully transferred to a new tube.

Subsequently, the purified bMDA products were fragmented to a peak size of 200 bp using an S220 Focused-ultrasonicator (Covaris) according to the manufacturer's instructions. Since, the volume of the pooled and purified bMDA products exceeded the volume of the tube for fragmentation, we repeatedly used the same microtube for the same bMDA pool. The fragmented DNA was purified using 1.8× volume of SPRI beads, and the washed beads were eluted with 1× volume of water. Although the concentration of the eluted DNA was similar, the fragmented DNA no longer increased the viscosity of the solution significantly, and the SPRI beads were magnetically separated. In the subsequent purification step, the eluate was transferred to a new tube carefully to prevent the transfer of SPRI beads. The contamination by SPRI beads may result in a non-specifically purified DNA fragments, leading to an increased number of non-barcoded DNA fragments post biotin purification step. A Qubit dsDNA Assay Kit (Invitrogen) was used per bMDA-seq procedure to quantify the DNA mass.

### Purification of biotinylated DNA using streptavidin coated beads

Twenty microliters of Dynabeads™ MyOne™ Streptavidin T1 (Invitrogen) beads were washed thrice with 1× wash buffer composed of 5 mM Tris-HCl (pH 7.5), 0.5 mM EDTA, 1 M NaCl, and 0.05% Tween 20. The washed beads were dissolved in a desired volume of 2× binding and washing (B&W) buffer, composed of 10 mM Tris-HCl (pH 7.5), 1 mM EDTA, and 2 M NaCl. An equal volume of the fragmented and purified bMDA products was added to make the final concentration of NaCl was 1 M. The mixture was then incubated for 15 min at room temperature (RT) with 20 rpm rotation to bind the biotinylated DNA to the streptavidin beads. The total volume of biotin binding could be increased up to 1 ml without a significant loss of biotin-binding capability. Next, biotin-bound beads were washed twice with 1× wash buffer and once with 1× B&W buffer. During the washing step, the suspended beads were incubated for 5 min at RT to increase the washing performance. Finally, the washed beads were resuspended in 100 µl of biotin elution buffer consisting of 10 mM Tris-HCl (pH 8.0), 1 mM EDTA, 1% SDS, and 0.73 mM D-Biotin. After incubation at 54 °C for 1 h with 800 rpm shaking, eluted biotinylated dsDNA was purified using MinElute PCR Purification Kit (QIAGEN) or SPRI beads. The biotin purification procedure was repeated to increase the specificity of purification.

### Sequencing library preparation

A conventional ligation-based Illumina sequencing library was prepared using a KAPA HyperPrep Kit (KK8500). After end-repair and A-tailing according to the manufacturer's instructions, the adapter ligation was performed at 20 °C for 8 h using 15 µM of adapter (TruSeq DNA Single Indexes Set A, Illumina). The ligation time was increased to 8 h because the presence of the 5′ biotin modification at one end of the DNA fragments can cause steric hindrance in the ligation reaction. The prepared library was quantified using a Qubit dsDNA Assay Kit (Invitrogen) and an Agilent 2200 TapeStation System or Agilent 2100 Bioanalyzer System.

### Target enrichment of the prepared library

A precapture pooling strategy was adopted to reduce the cost of target capture. Target enrichment of the bMDA library was performed using the SureSelectXT2 Reagent Kit (G9621A, Agilent) or SureSelect XT HS2 DNA Reagent Kit (G9981A, Agilent), as per the manufacturer's instructions, except for a few modifications that were suitable for the structure of the bMDA library. We omitted the library preparation procedure of the SureSelect system because the library had already been prepared. The SureSelect XT2 Blocking Mix was used to passivate the Illumina adapter sequences of the bMDA library. For hybridization, SureSelectXT Focused Exome probes (5190-7787, Agilent) (17.8 Mb) were used to study the FPR and ADO rates of HL-60 single-cells, and

SureSelect Human All Exon V5 probes (Agilent) (50.4 Mb) were used to study SNV in TNBC tissue sections. A single reaction volume of the capture probe was used to capture eight and four bMDA products for the respective probe products, which reduced the number of target enrichment reactions by eight and four folds, respectively. Even with the reduced probe-to-sample ratio, we could cover target regions at 250× on average per sample. Thus, to capture a pool of 48-plex bMDA products, we required 6 and 12 target enrichment reagents and probes for the respective probe types.

To analyze somatic SNV from TNBC samples, we arbitrarily selected 24 out of 28 samples from T2 and 8 out of 20 samples from T1.

**Tn5-based single-cell genome sequencing library preparation**
A Tn5-based single-cell DNA sequencing library was prepared as described previously[48,49]. To prepare Tn5 transposase, a pTXB1 cloning vector with hyperactive E54K and L372P mutations was obtained from Addgene and transformed into DH5a cells (NEB). After cell culturing and IPTG induction, sonication was carried out to lyse the cells. To get purified Tn5 transposases, a chitin column was used with the supernatant from the lysed cells and Tn5 transposase was dialyzed by a dialysis buffer, which is composed of 100 mM HEPES-KOH (pH 7.2), 0.2 M NaCl, 0.2 mM EDTA, 2 mM DTT, 20 % glycerol, 0.2 % Triton X-100. Then, transposome was assembled with the purified Tn5 transposase and preannealed mosaic end double-stranded oligonucleotides. We isolated a single SK-BR−3 cell line for the Tn5-based single-cell genome analysis. The isolated cell was lysed using proteinase K (P4850, Sigma-Aldrich) at 50 °C for 1 h. The lysate was then mixed with Tn5 transposase and incubated at 55 °C for 20 min, and the tagmentation process was halted by incubation at 37 °C for 1 h with proteinase K. The tagged DNA fragments were subjected to PCR amplification for 13 cycles, followed by purification and a second round of PCR for 7 cycles. The second round of PCR were monitored in real-time to quantify the library conversion efficiency resulting from the Tn5 tagmentation process.

**Sequencing and data pre-processing**
Illumina MiniSeq, HiSeq 2500, Hiseq 4000, or NovaSeq 6000 systems were used for short-read sequencing. The generated FASTQ file was first demultiplexed by comparing the first six base calls of read 2 with the bMDA barcodes. After demultiplexing, the bMDA barcode sequence was removed from FASTQ. Reads were then aligned to the human reference genome (GRCh37) using BWA-MEM (v0.7.15)[50] with default parameters, and the resulting Sequence Alignment Map (SAM) file was sorted by chromosomal coordinates using SAMtools (v1.11)[51]. Subsequently, the Picard Toolkit (v2.9.2) (http://broadinstitute.github.io/picard) MarkDuplicates was used to identify duplicated reads, and the Genome Analysis Toolkit (GATK v3.7-0)[52] was used to perform local realignment around indels and base quality score recalibration (BQSR). Finally, reads with a mapping quality less than 30 or that were artificially generated by supplementary alignment were removed to produce an analysis-ready BAM file.

**CNA and Lorenz curve analysis**
The HL-60 cell line was used to evaluate the technical performance of bMDA. bMDA and conventional MDA were performed using either 66 pg of gDNA or a single cell as the template. In single-cell experiments, large cell-to-cell variations are usually detected either by cellular variations in lysis efficiency or stochastic MDA amplification bias. We performed MDA from 66 pg of gDNA to prevent cellular variation from affecting performance evaluation. Data obtained from formalin-fixed paraffin-embedded (FFPE)-treated HL-60 cells were used as representative of a low-quality sample.

CNA and Lorenz curve analyses were performed based on shallow-depth whole-genome sequencing data. For each sample, ~1 million NGS reads were generated for CNA analysis and 0.24 million reads were generated for the Lorenz curve analysis. We used variable-sized binning methods[3] for both analyses. Briefly, the human genome was split into 10 k (median bin size = 276 kb) or 15 k (median = 184 kb) bins, where each bin size was adjusted to have an equal expected number of uniquely aligned reads. After NGS reads were assigned to each bin, the read counts of each bin were normalized by GC content using LOWESS to correct the GC bias. Assuming that the samples to be analyzed are near-diploid, the GC-normalized read densities of each bin were scaled to have an average value of 2. Then, CNA events of a single sample were detected by circular binary segmentation (CBS) of the Bioconductor DNACopy package[53]. The detected segments were processed using MergeLevels[54] to merge segments whose difference was not significant for a group of samples amplified from the same cell line or from the same patients with cancer. The parameters used for the CBS to perform the MergeLevels were alpha = $10^{-10}$, nperm = 1000, min. width = 5, undo.SD = 0.8., and Ansari. sign = 0.1.

For Lorenz curve analysis, NGS reads were assigned to each variable-sized bin and Lorenz curve was then constructed by accumulating the fraction of reads assigned to each bin. Traditionally, the Lorenz curve is generated using high-depth whole-genome sequencing data to visualize read depth bias at the base-pair (bp) resolution. In our study, however, we specifically aimed to evaluate the efficacy of bMDA in detecting copy number alterations (CNAs) through the use of low depth whole-genome sequencing data, while simultaneously utilizing high-depth targeted sequencing for the detection of SNVs.

**SNV detection**
To detect SNV, 150 paired-end (PE) sequencing was performed to generate a 1.5 Gb/sample for the focused exome captured library and 10 Gb/sample for the library captured using the SureSelect Human All Exon V5 probe. The sequencing amount was determined to cover the target region with a coverage depth of at least 100× for each bMDA product.

Three different variant callers were used for reliable detection of somatic SNVs in TNBC samples, as described previously[4,55]. First, all bMDA and tumor bulk data were processed simultaneously using GATK UnifiedGenotyper with default parameters. The GATK VariantRecalibrator was used for variant quality score recalibration (VQSR). Variant annotations used for the VQSR training included DP, QD, MQ, FS, MQRankSum, and ReadPosRankSum; and highly validated variant resources including HapMap 3.3, Omni 2.5 M, 1000 G phase1, and dbSNP build 137 were used for the training database. Variants with a quality score of less than 50 or germline variants detected by the matched normal data were removed to obtain GATK somatic variants. We used BEDTools (v2.26.0)[56] for the filtration of germline variants. VarScan2 (v2.3.9)[57] and MuTect (v1.1.4)[58] were used with default parameters to obtain VarScan and MuTect somatic variants, respectively.

For each sample, variants called by at least two different callers were considered high-quality variant calls of the sample that were less susceptible to false-positive mutation calls, which usually arise when using an individual caller. To remove false-positive mutations caused by MDA or bMDA amplification errors, high-quality variants detected in at least two samples were selected to obtain confident variant sites across all samples. Since the FPR of bMDA was ~$10^{-5}$/base, the theoretical possibility of detecting the same false-positive mutations on the same site of two different samples is ~$10^{-10}$, which is almost negligible or likely to be lower than the value because there are three possible alternate bases that vary from the reference base. Among the confident variant sites, a somatic variant of a sample is called if (1) the confident variant is called by one of three different callers, or (2) the variant allele count of the loci is significantly larger than the count of other non-reference bases occurring by sequencing errors (Fisher's exact test, $p < 10^{-3}$). Both, NGS reads with mapping quality and base quality score greater than 30 were used for base counting for the significance test based on the read depth.

### ADO rate and false-positive mutation detection rates (FPR) calculation

To calculate the ADO rate, confident heterozygous SNPs were detected using bulk sequencing data. GATK UnifiedGenotyper, VarScan2 mpileup2snp, and MuTect2 (GATK v4.1.9.0)[58] were used with default parameters to obtain the intersection of the SNPs detected by the three different callers (triple-called SNPs). Subsequently, heterozygous variants detected in the genome region with copy number 2 were regarded as confident heterozygous SNPs. Allelic dropout events were counted if single-cell bMDA data had coverage ≥10, but one of the two alleles was missing in the confident heterozygous SNP sites.

To calculate the FPR, confident homozygous sites were first determined from bulk sequencing data. Among the genome regions with coverage ≥20, a genome position was regarded as a confident homozygous site if only one type of allele was detected at the site. Varscan2 somatic was used with single-cell bMDA data as the tumor pile-up input and bulk sequencing data as the normal pair input. Somatic mutations detected by variant calls were used to determine false-positive mutation detection events.

To assess the sensitivity and FPR of SNV calling in cell cluster of TNBC tissue sections, we applied same pipeline for tumor cell clusters and compared it with germline mutations detected in normal bulk sequencing data.

### Inference of the tumor purity

Tumor purity was inferred by feeding somatic single nucleotide variants (SNVs) and the segment mean of copy number alteration (CNA) detection results into the ABSOLUTE algorithm[59]. The sequencing data from each tumor subclone (c2–c5) were merged and processed together to obtain representative tumor purity values for each subclone.

### Inference of the tumor phylogeny

To infer the phylogenetic tree based on SNV, a binary variant matrix across samples was constructed from the detected somatic variants by providing a value of 1 if the variant was detected, NA if coverage was <5 and the variant was not detected at the locus, and 0 if the variant was not detected. Subsequently, pairwise distances were calculated using Manhattan distance, and a phylogenetic tree was inferred employing a balanced minimum evolution algorithm[60] using the R package ape (v5.5)[61]. The inferred tree was rooted in an artificially generated diploid node, which had zero values for each variant site. All trees were constructed using the ggtree[62] software package.

### Detection of SV and kataegis

We conducted whole-genome sequencing (WGS) of 8 Gb per bMDA sample to analyze SVs and Kataegis. WGS of 400 Gb (120×) was also performed for analyzing tumor bulk gDNA and 100 Gb (30×) for analyzing matched normal gDNA. DELLY (v0.9.1)[63], Manta (v1.6.0)[64], and GRIDSS2 (v2.12.2)[65] were used with the default parameters for detecting somatic SVs. First, somatic SV detected by MANTA and highly confident somatic SV detected by GRIDSS2 were merged to obtain highly confident somatic SV for each sample. The detected SVs were merged across all bMDA samples and tumor bulk data, followed by the removal of potentially false variants detected in the genomic region where the mapping quality of each NGS read was less than 20. Variants detected by more than two different samples were retained to obtain highly confident somatic SV across all samples.

To increase the sensitivity of SV detection, we also obtained low-confidence somatic SVs by combining the variants detected by DELLY2, Manta, or GRIDSS2. After combining the low-confidence variants across all samples, variants detected at low-mapping-quality genomic regions were removed. Finally, SV that were detected in the tumor bulk data and those that were detected in at least two different samples were retained to obtain bulk-supported highly confident SVs.

The highly confident somatic SV and bulk-supported highly confident somatic SV were combined to obtain the final detected SVs. We used SURVIVOR (v1.0.7)[66] to filter or merge the SV. We used AnnotSV (v3.1.2)[67] to annotate the detected SV, Circos (v0.69-9)[68] to visualize the detected translocations, and ChromoMap (v4.1.1)[69] to visualize the detected SV on the chromosomal ideogram.

To detect Kataegis events, somatic SNV across the whole genome were first called using GATK UnifiedGenotyper with default parameters, and all bMDA, tumor bulk, and matched normal sequencing data were processed simultaneously to obtain a single VCF file. After removal of germline variants that were detected in matched normal data, SNV that were detected by at least six different samples with more than 20 total supporting reads for the variant were regarded as highly confident somatic SNV. Through this filtration, we could avoid false-positive mutation detection that can arise from the MDA process. Subsequently, Kataegis Portal (v1.0.3)[70] was used to detect Kataegis events.

### Spatial tumor evolution inference

To visualize the integrative genomic profile over time and space, we first analyzed subclone-specific mutations for single nucleotide variants (SNVs), structural variants (SVs), and Kataegis (Supplementary Data 1). For SNVs and Kataegis, we considered the presence of a mutation based on the majority voting among the spatial microniches, with the most prevalent mutation profile representing the mutational profile of a specific subclone. For SVs, a mutation was considered present if it was detected in at least two samples belong to either the subclone or tumor bulk. Overlapping mutations between different subclones was regarded as an evidence for the presence of a common ancestor, suggesting shared evolutionary origins. Subclone-specific mutations were considered as evolutionary changes specific to each subclone over time, reflecting the progressive development and diversification of subclones. To prioritize the genes to be displayed, we employed the following criteria: (1) deleterious protein-altering mutations based on the SIFT score, and (2) consideration of genes known to be associated with breast cancer, as supported by a literature search.

### Statistics and reproducibility

For the validation of the bMDA technology, all experiments were conducted for at least in triplicate for each experimental group. The mean, standard deviation, and standard error of the mean were computed to facilitate comparisons between the different groups. Statistical analysis was carried out using appropriate methods, including the Wilcoxon rank-sum test, Student's *t* test, and one-way ANOVA, as required, to assess the significance of differences between the groups. In the assessment of the applicability of bMDA in TNBC tissue sections, two different TNBC tissue sections were evaluated. Sample sizes were chosen based on the experimental context, or to demonstrate the potential use of the developed technology, and no statistical method was used to predetermine the sample size. All attempts at replication, including both successful and unsuccessful ones, are transparently presented in the figure plots and no data was excluded from all analyses. Breast cancer patients were randomly selected and the investigators were not blinded to allocation during experiments and outcome assessment.

### Reporting summary

Further information on research design is available in the Nature Portfolio Reporting Summary linked to this article.

## Data availability

All sequencing data generated in this study have been deposited in the NCBI Sequence Read Archive under accession code PRJNA986002. The GRCh37 human genome reference is available for download from Ensembl (https://grch37.ensembl.org/info/data/ftp/index.html). Source data are provided with this paper.

## Code availability

All custom scripts are available on GitHub (https://github.com/BiNEL-SNU/bMDA)[71].

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

## Acknowledgements

This work was supported by the BK21 FOUR program of the Education and Research Program for Future ICT Pioneers, Seoul National University in 2023 (awarded to S. Kwon, S.L., N.K., and K.S.), the Ministry of Science and ICT (MSIT) of the Republic of Korea and the National Research Foundation of Korea (NRF-2020R1A3B3079653 awarded to S. Kwon and 2020M3H1A1073304 awarded to A.C.L.), and the Industrial Strategic Technology Development Program (20024391, Development of a Personalized Breast Cancer Diagnosis and Treatment Technology-Product Using Single-Cell Multi-Omics Analysis and In Situ Sequencing of Circulating Tumor Cells) funded By the Ministry of Trade, Industry & Energy (MOTIE, Korea) (awarded to A.C.L.).

## Author contributions

S. Kwon, H.L., W.H., J.K., A.C.L., and S. Kim conceived and designed the study. J.K. and S. Kim developed the bMDA method. J.K., S. Kim, A.C.L., H.Y., A.C., and H.K. performed bMDA experiments. T.R., O.K., and Y.J. performed the Tn5 experiments. H.L. and W.H. provided the breast cancer tissue samples and managed the IRBs. H.L. and J.Y.K. prepared the tissue sample. J.K. analyzed the data with the help of K.S., S.B., and N.K. H.S.R. performed pathological inspections. J.K., A.C.L., and S. Kwon wrote the initial manuscript. S. Kim, H.L., S.W.S., S.B., K.S., A.C., T.R., Y.C., H.K., O.K., Y.J., and S.L. reviewed and edited the manuscript.

## Competing interests

H.L. and W.H. report being members of the board of directors and holding stock and ownership interests at DCGen, Co., Ltd., which is not relevant to this study. S.L., A.C.L., and S. Kwon hold share in Meteor Biotech, Co. Ltd. S. Kwon, J.K., A.C.L., S. Kim, and A.C. have filed patent related to this manuscript (PCT/KR2022/008690). The remaining authors declare no competing interests.
