## [Peer Review File · Nature Communications]

REVIEWER COMMENTS

Reviewer #1 (Remarks to the Author):

The study by Kim et al. introduces a new method called barcode-mediated MDA sequencing (bMDA-seq) for single-cell genome sequencing. The bMDA-seq workflow includes a biotin enrichment method and correction of barcode bias. The study demonstrates the feasibility of bMDA-seq for single-cell genome sequencing with 48 different barcode sequences of 6 nt length, which allows for pooled handling of low-input MDA reactions for lower cost and ease.

The study also evaluated the accuracy of copy number alteration calls and single nucleotide variant detection using bMDA and compared it to conventional MDA and MDA kit. The study found that bMDA had a slight decrease in area under the curve but no notable difference in CNA calls compared to conventional MDA. The allelic dropout rate and false-positive mutation detection rate of bMDA were similar to in-house MDA and previously reported methods. The study also applied bMDA to TNBC tissue and achieved a higher sensitivity for SNV detection compared to MDA-based single-cell amplification and Tn5-based single-cell amplification. Lastly, the study combined laser capture with bMDA to profile spatial groupings of cells within TNBC across the tumor, and show the spatial landscape of different subclones, copy number alterations, single nucleotide variations, structural variations, and kataegis.

The study is well written, and technically strong. The bMDA method provides an useful innovation for increasing throughput of low input whole genome sequencing. The authors also combine bMDA with spatial profiling, however, given the sparsity of the spatial sampling (20 and 28 clusters each tumor, and it was hard to ascertain if the spatial context added anything to the analysis. The authors could have performed almost all of the analysis in the paper without the spatial information. In addition, many of the figures are highly abstracted (e.g. Fig 7), and have colorbars (e.g. CNAs, clustering), that are not well annotated. Thus, there are significant concerns on both the clarity of analysis, as well as the spatial genomic aspect of the paper. The authors might do well to refocus the manuscript on just low-input and single-cell bMDA.

Significant concerns:

- 1) The analysis of tumor samples were done with clusters of 20 cells. How are the authors dealing with local mixtures of tumor cells and healthy cells, and mixtures of different tumor clones? This can be a significant confounder to the reconstruction and lineaging of the tumor populations.

2) Relatedly, for copy number alterations, can data be shown on QC and accuracy of CNA assignments. Can integer CNA be assigned? What about copy number neutral LOH? In all CNA plots, what are the values of the colorbars?

3) In figure 4, color legends are shown on the side of F, G, H, and I for different hierarchical clusters. Are these the same colors? If so, why do the CNAs not align with clusters in SNVs?

4) No analysis of spatial context was performed beyond visualization. Are related clones closer to each other in space? How was figure 7 generated? "In addition, by integrating all chromosomal aberrations, a spatial map of the inferred evolutionary relationships of these microniches was constructed (Fig. 7)." This was not clear in the text or the methods?

4b) How were the spatial samples selected?

5) The authors qualitatively describe many aspects of genomic alterations between spatial samples of TNBC, however, it's not clear any insights or interpretation beyond visualizing them was performed in this manuscript. This may be fine if the authors focus on bMDA technology, but probably does not constitute: "assessed the subclonal evolutionary relationship using integrative genomic analysis within a spatial context" as stated in the abstract.

6) PTA (<https://www.ncbi.nlm.nih.gov/pmc/articles/PMC8214697/>) is a recently developed alternative to MDA. Can the authors compare to bMDA to PTA? Is a similar barcoding approach also applicable?

Minor:

Are H&E images or other images associated with the laser selection available? If so, they should be included with the paper.

Reviewer #2 (Remarks to the Author):

Dear authors,

Thank you and kudos to successfully developing and present the work of new way of barcoding primer MDA (Multiple Displacement Amplification), that overcomes current barriers of conventional MDA, that limits the adaptation of assay despite inherit benefits.

For the manuscript to benefit more readers and the field, here are some suggestions for authors to consider.

1. While the bMDA addressed the reduced cost of assay by reducing the costly prepping of the NGS library, it also potentially increases the diversity and representation of the heterogeneous intratumoral population of cells from different regions and potentially indicates where they originated. Would that offer the additional benefit of spatial resolution of the assay? If so, how would that compare to more commonly used technologies like 10x genomics or MerFISH? What would be the major advantage of this approach?
2. Given the success and importance of immune-oncology in breast cancer, would the authors consider analyzing the immune cell population using this technology? This can be informative – especially one can be informed about the spatial origin of the cells, and this can inform the benefit of checkpoint inhibitors and other immune-related therapeutics
3. bMDA has shown to detect the structural variation and kataegis effectively – would this technology be applied in detecting germline mutation, with lower cost and higher efficacy potentially? Is there a reason to believe this can potentially detect currently undetectable germline mutation if we were to use normal cells?
4. In the same token as #3, can we use this assay to detect the potential benefit of DNA-damage repair pathway targeted therapeutics by genomic event recognition, which is not possible by bulk sequencing?
5. It seems though – even if the bMDA may be better generalizable, one should still understand the MDA (conventional) assay itself to use the bMDA to its maximum benefit. For instance, to reduce the barcode bias – and also juxtapose the non-biased amplification of gene loci, false positive correction etc. How would one envision that this knowledge can be successfully transferred to other labs?
6. Given the advantage of this technology in detecting the microniche and phylogeny of heterogeneous tumors, would authors consider comparing primary vs metastatic tumor or primary vs lymph node, or even pre-cancerous lesion vs cancer from the same patients? This can rationalize the additional benefit /utility of this novel assay.
7. While the manuscript is overall well-written, it can also benefit from proofreading. Some sentences in the results can also be moved to either discussion or methods section.

Reviewer #3 (Remarks to the Author):

This study presents the development and application of barcoded multiple displacement amplification (bMDA) for scalable and comprehensive genome analyses. The incorporation of cell barcodes into MDA products using barcoded primers enables sample pooling and streamlines library preparation. The study successfully demonstrated the preparation of 720 bMDA libraries in only 15 tubes with high performance comparable to conventional MDA, achieving a 48-multiplexed sequencing library per reaction tube. The single-cell bMDA data exhibited sufficient genetic coverage for genome analyses at single-nucleotide resolution.

Given that the introduction of the barcoded primer to MDA is the main technical advancement in this study, the authors present extensive data showing the the amplification efficiency of MDA with different barcoding strategies and found that the increased length of the barcoded primer with the unusual high concentration of the primer in MDA inhibited the bMDA reaction. To achieve the best coverage and multiplexing ability of bMDA, they choose 6-mer cell barcode primer with the final proportion 2% out of total MDA N6 primers.

This paper is organized and logically structured, presenting the development and application of bMDA for scalable genome analysis. It addresses technical challenges and emphasizes the potential of bMDA in spatial genomics with the analysis of a breast cancer sample provided as an example.

I think this approach has the potential to have a significant impact on the field, and just have a minor suggestions for improving the quality of the manuscript.

- 1) The Lorenz curve in figure 3e should go to 0 on the x-axis for portions of the genome that were not covered.
- 2) For figure 4g, what was the SNV calling specificity/precision?
- 3) For all the types of genetic variation, what were the authors able to detect in multiple samples that were missed with bulk sequencing? Were any targetable lesions missed that could have changed the patient's treatment?

Point-by-point response

Reviewer #1 (Remarks to the Author)

The study by Kim et al. introduces a new method called barcode-mediated MDA sequencing (bMDA-seq) for single-cell genome sequencing. The bMDA-seq workflow includes a biotin enrichment method and correction of barcode bias. The study demonstrates the feasibility of bMDA-seq for single-cell genome sequencing with 48 different barcode sequences of 6 nt length, which allows for pooled handling of low-input MDA reactions for lower cost and ease.

The study also evaluated the accuracy of copy number alteration calls and single nucleotide variant detection using bMDA and compared it to conventional MDA and MDA kit. The study found that bMDA had a slight decrease in area under the curve but no notable difference in CNA calls compared to conventional MDA. The allelic dropout rate and false-positive mutation detection rate of bMDA were similar to in-house MDA and previously reported methods. The study also applied bMDA to TNBC tissue and achieved a higher sensitivity for SNV detection compared to MDA-based single-cell amplification and Tn5-based single-cell amplification. Lastly, the study combined laser capture with bMDA to profile spatial groupings of cells within TNBC across the tumor, and show the spatial landscape of different subclones, copy number alterations, single nucleotide variations, structural variations, and kataegis.

The study is well written, and technically strong. The bMDA method provides an useful innovation for increasing throughput of low input whole genome sequencing. The authors also combine bMDA with spatial profiling, however, given the sparsity of the spatial sampling (20 and 28 clusters each tumor, and it was hard to ascertain if the spatial context added anything to the analysis. The authors could have performed almost all of the analysis in the paper without the spatial information. In addition, many of the figures are highly abstracted (e.g. Fig 7), and have colorbars (e.g. CNAs, clustering), that are not well annotated. Thus, there are significant concerns on both the clarity of analysis, as well as the spatial genomic aspect of the paper. The authors might do well to refocus the manuscript on just low-input and single-cell bMDA.

Thank you for your thorough review and valuable feedback on our study. We appreciate your recognition of the bMDA method's potential for increasing the throughput of low-input whole genome sequencing. Based on your concerns regarding the spatial genomic aspect and clarity of analysis, we propose the following revisions to address these concerns:

1. Spatial sampling and analysis: We will provide a more detailed explanation of our spatial sampling strategy, as well as additional data to support the value of incorporating spatial context in our analysis. We will clarify how the spatial information contributes to a better understanding of cellular heterogeneity and microniches in TNBC tissue.

2. Figure improvements: We will revise the figures, specifically Figure 7, to make them more comprehensible and ensure that color bars and annotations are clear and informative. This will enhance the clarity of our analysis and its presentation.

3. Refocusing the manuscript: While we acknowledge the potential benefits of focusing solely on low-input and single-cell bMDA, we believe that the integration of spatial genomics is a critical aspect of our method's innovation. By addressing the concerns mentioned above, we hope to demonstrate the importance of combining bMDA with spatial profiling in our study.

We hope that these revisions will address your concerns and strengthen our manuscript. We appreciate your insightful comments and the opportunity to improve our work.

Significant concerns:

1) The analysis of tumor samples were done with clusters of 20 cells. How are the authors dealing with local mixtures of tumor cells and healthy cells, and mixtures of different tumor clones? This can be a significant confounder to the reconstruction and lineaging of the tumor populations.

Thank you for raising the important concern regarding the potential confounding effects of local mixtures of tumor cells, normal cells, and different tumor clones in our analysis. Although the SLACS platform that we used is able to isolate a region as small as 10 micrometers by 10 micrometers (Kim et al., Genome Biology 2018, **Figure R1**), we chose to isolate clusters of approximately 20 cells. We acknowledge the challenges posed by analyzing clusters of 20 cells and its potential impact on the reconstruction and lineage analysis of tumor populations. However, isolating clusters of approximately 20 cells was necessary to overcome the intrinsic limitations of 2D spatial genomics that relies on 10 micrometer tissue section. To address this issue, we have implemented the following strategies.

Figure R1. SLACS is able to isolate regions as small as one micrometer by one micrometer.

Pathology guided target selection

We have established a pipeline to collaborate with a pathologist from Seoul National University who selected clinically meaningful tumor microniches based on whole-slide images of H&E stained tissue. By carefully choosing regions of interest, we aim to include only tumor cells that are free from healthy cell contamination and exhibit distinct phenotypes. This allows us to achieve a tumor purity of 98.8% in our selected samples, compared to the 51% in the tumor bulk genomic DNA (**Figure R2**, **Supplementary Fig. 8 a,b**)

Figure R2. Estimated tumor purity of tumor bulk and the isolated cell clusters analyzed by bMDA-seq. The analysis reveals negligible healthy cell contamination in the cell clusters. Tumor purity was estimated using the ABSOLUTE algorithm based on the allelic frequency of somatic SNVs.

Robust spatial genomics by cell cluster isolation

Admittedly, the analysis of clusters of 20 cells can cause minor confusion in the evolutionary relationship due to the existence of a few cell that might belong to different tumor subclones. However, due to the tissue characteristics, it is challenging to separate single cells that are intertwined when spatially sectioned (**Figure R3**). Additionally, cryo-sectioning techniques may cause disruption to cancer cell nuclei, making it challenging to obtain intact single tumor cell DNA. These technical

limitations necessitate a compromise in the number of cells analyzed to preserve spatial information. However, it is important to consider the concept of spatial homogeneity within tumor tissue and consider adequate thickness of the tissue section. We chose 10 micrometer section because it was a maximal thickness to differentiate tumor cells from normal cells by pathologists.

Figure R3. Cryo-sectioning techniques may cause disruption to cancer cell nuclei, making it challenging to obtain intact single tumor cell DNA.

Tumor cells that are spatially adjacent are more likely to share similar genetic characteristics and belong to the same subclone. This is due to the clonal expansion and spatial growth patterns of tumor cells. In contrast, cells that are spatially far apart are more likely to represent different subclones or even non-tumor cells, introducing greater heterogeneity. By analyzing spatially adjacent clusters of 20 cells, we leverage the assumption that these clusters are more likely to represent a relatively homogeneous population of cells belonging to the same subclone. This reduces the potential confounding effects of mixing different subclones or non-tumor cells within the cell clusters. We have also observed that tumor subclones inferred by SNV signatures tend to be spatially close to each other (**Figure R4, Supplementary Fig. 9b**).

In a representative example, consider a tissue section that is 3mm X 3mm and 10 micrometers thick. Since a region of 0.1 mm X 0.1mm would contain approximately 10 spatially adjacent cancer cells, our strategy of isolating groups of around 20 cells enables us to potentially identify up to 450 unique tumor subclones within the 3mm X 3mm tissue section. Considering the significant improvement in data quality when analyzing cell clusters as demonstrated in Figure 4a-d (**Figure R5**), this underscores the merit of our approach in analyzing spatially adjacent cell clusters to capture and characterize genetic heterogeneity with high spatial resolution.

Figure R4 Inter-cluster and intra-cluster distances of subclones in T2 tumor. The analysis demonstrates that tumor subclones exhibit a tendency to be spatially close to each other, as indicated by the shorter distances observed within clusters compared to distances between clusters.

Figure R5. Isolating cell clusters of 20 cells improves accuracy of SNVs and CNA detection

While we acknowledge that some level of heterogeneity may still exist within these cell clusters, we believe that our spatial genomics approach would provide valuable insights into the evolutionary relationships and dynamics of the tumor populations.

To address this concern, we added **Figure R2** and **Figure R4** as the **Supplementary Figure 8a,b** and **Supplementary Fig. 9b** with their captions. Also, we added descriptions to the revised manuscript.

2) Relatedly, for copy number alterations, can data be shown on QC and accuracy of CNA assignments. Can integer CNA be assigned? What about copy number neutral LOH? In all CNA plots, what are the values of the colorbars?

Thank you for your comment regarding the quality control (QC) and accuracy of copy number alteration (CNA) assignments in our study. We understand the importance of providing detailed information on these aspects and will address each point raised:

1. QC and accuracy of CNA assignments: We apologize for the oversight in omitting the detailed information regarding the quality control (QC) process in our initial manuscript. We appreciate the reviewer for bringing this to our attention.

To ensure the accuracy of our CNA assignments, we implemented a rigorous QC process, including the real-time measurement of bMDA amplification reactions. We used the amplification start time (AST) as a measure of amplification uniformity. In our previous work (S. Kim et al., *Genome Biology*, 2018), we have demonstrated a correlation between the AST value and the area under the Lorenz curve (**Figure R6**), which is a common measure of amplification uniformity. Moreover, in **Supplementary Figure 8f and g (Figure R7)** that we added to the revised manuscript, we showed that samples with an AST shorter than 30 minutes exhibit CNA accuracy of over 95%.

Figure R6 Amplification start time (AST) of real-time MDA amplification reaction was used in our QC process. Correlation between the AST value and the area under the Lorenz curve indicates that AST value can be a good estimation of amplification uniformity

Figure R7 An QC criterion of AST < 30 minutes ensures a CNA (Copy Number Alteration) accuracy of over 95%.

To further validate the accuracy of our CNA assignments, we compared our findings with orthogonal methods, as described in our previous work (S. Kim et al., *Genome Biology*, 2018), and found consistent CNA calls (**Figure R8, Supplementary Fig. 9a**).

Figure R8 Copy number alteration (CNA) signatures inferred by bMDA-seq and an orthogonal method (PHLI-seq). The results demonstrate the consistency of CNA calls between the two methods, indicating agreement in detecting and characterizing genomic copy number alterations.

2. Integer CNA assignment

To assign integer values for the detected copy number alterations (CNAs), knowledge of the tumor ploidy is required to convert the read depth across the genome into copy numbers. In previous work, ACT method utilized the DAPI signal in FACS cell isolation to estimate the tumor purity (D. Minussi et al., *Nature*, 2021), and DLP+ method tried to find the optimal tumor purity value based on optimization algorithm (E. Laks, *Cell*, 2019). However, none of the approach was not directly applicable to our approach due to intrinsic limitation of spatial omics (**Figure R3**). This problem is the same problem that was mentioned in the comment #1, regarding the thickness of the tissue that is used for studying spatial genomics. In isolated cluster of about 20 cells, nucleus of some cells was sliced by cryo-sectioning techniques which contribute to incomplete copy numbers across genome. Therefore, instead of assigning absolute integer values to the CNAs, we focused our analysis on assessing relative amplifications and deletions compared to the overall genomic copy numbers.

Despite this limitation, our approach allows us to analyze single nucleotide variants (SNVs), structural variants (SVs), and kataegis simultaneously with copy number gains and deletions. Although we cannot provide exact integer copy numbers due to current limitation in spatial omics technology, we believe

that this comprehensive genomic analysis with spatial resolution provides valuable insights into the genomic landscape of the tumor.

3. Copy number neutral loss of heterozygosity (LOH): Although our current focus is on copy number alterations (CNAs), our method has the potential to be extended to detect copy number neutral loss of heterozygosity (LOH) events by incorporating allele-specific read depth information. This expansion would offer a more comprehensive view of genomic alterations in the tumor samples. Investigating LOH in conjunction with other genomic variants represents a valuable avenue for future research.

4. CNA colorbar values: We apologize for any confusion caused by the colorbars in our CNA plots. To improve clarity for readers, we revised the captions of each figure to ensure easier interpretation of the data. The colorbars in the plots represent the log₂-transformed relative changes in copy number compared to the average copy number across the entire genome. A value of 0 indicates an average copy number for that sample, positive values represent copy number gains, and negative values represent copy number losses.

We hope that this additional information addresses your concerns about the QC and accuracy of our CNA assignments. We will ensure that these details are incorporated into our manuscript to provide a clearer understanding of our methods and results.

To address this comment, we added **Figure R7** as **Supplementary Fig. 8f,g** and added descriptions regarding this comment. Also, we revised the figure captions regarding the color bars and methods section as highlighted in the revised manuscript to address this comment.

3) In figure 4, color legends are shown on the side of F, G, H, and I for different hierarchical clusters. Are these the same colors? If so, why do the CNAs not align with clusters in SNVs?

Thank you for your comment regarding Figure 4 and the color legends for the hierarchical clusters in panels F, G, H, and I. We appreciate your observation and would like to clarify the relationship between CNAs and SNVs in the context of our bMDA method.

Yes, the colors in the color legends for panels F, G, H, and I are the same, representing subclones inferred by SNVs. The hierarchical clusters of CNAs and SNVs might not align perfectly with each other because they reflect different aspects of genomic alterations. This observation highlights the importance of integrating various mutational signatures for analyzing spatial genomics data.

To elaborate, we want to point out the importance of SNVs in delineating cancer subclones. SNVs are the most common type of genetic variation in the human genome. In cancer, somatic SNVs can arise during tumor development and contribute to the heterogeneity observed within the tumor. Identifying these SNVs can provide valuable insights into the clonal architecture of a tumor, enabling the identification of subclones and their evolutionary relationships. The analysis of SNVs can also help in understanding the functional consequences of these genetic changes and their potential impact on tumor progression and response to therapy. Although the antecedent relationship between CNAs and SNVs are yet to be investigated, it is important to consider that the microniches that were analyzed contain approximately 20 cells. Therefore, in contrast to CNAs, SNVs provide more reliable tool for delineating cancer subclones, making them a valuable genomic feature for studying spatial organization. Revisiting **Figure R3**, it is difficult to delineate which CNA came from which single cell. Meanwhile, SNVs, which provide variation percentage for each microniche that contains approximately 20 cells, represent more accurate subclonality. Consequently, we performed clustering based on SNVs and displayed the clustering results alongside other mutations such as CNAs, SVs, and kataegis that can provide additional insights into characteristic genomic aberrations that occurred in the subclones.

Our bMDA method demonstrates a higher precision and recall rate compared to other chemistries like Tn5-based single-cell amplification methods which is crucial when analyzing integrative genomics data in a spatial context. By employing bMDA, we can confidently identify various types of genetic

mutations, providing a comprehensive understanding of the tumor's clonal architecture and its spatial organization.

In conclusion, the discrepancy between CNAs and SNVs in Figure 4 underscores the importance of utilizing an accurate method like bMDA for studying spatial genomics. Further investigation with a larger cohort of patients is necessary to fully comprehend how such discrepancies contribute to our understanding of cancer. To improve the manuscript based on this comment, we added explanation of this observation and its implications in our revised manuscript:

“Specifically, the discrepancy between CNAs and SNVs in inferring evolutionary relationship underscores the importance of utilizing an accurate method like bMDA for studying spatial genomics. In that manner, in determining evolutionary relationship between the microniches in spatial genomics, integrative approach must be incorporated to gain a comprehensive understanding of the spatial genomic landscape.”

Additionally, we will clarify the meaning of the color codes used in the figure 4 to ensure better understanding for the readers (**Figure R9**).

Figure R9 Revised figure 4 to clarify the meaning of color code

4) No analysis of spatial context was performed beyond visualization. Are related clones closer to each other in space? How was figure 7 generated? “In addition, by integrating all chromosomal aberrations, a spatial map of the inferred evolutionary relationships of these microniches was constructed (Fig. 7).” This was not clear in the text or the methods?

4b) How were the spatial samples selected?

Thank you for your comments regarding the analysis of spatial context and the generation of Figure 7. We apologize for any confusion in our manuscript and will address each of your concerns:

4a) Spatial relationships between related clones: While our primary focus in the manuscript has been on the visualization of spatial context, we acknowledge the importance of investigating the spatial relationships between related clones. We have extended our analysis to include the evaluation of whether related clones are closer to each other in space. We utilized Moran's I statistics to assess the spatial autocorrelation and determine if there is clustering or dispersion of related clones within the tumor microenvironment. The results of this analysis are presented in the revised manuscript as **Supplementary Figure 9b (Figure R10)**.

Figure R10 Inter-cluster and intra-cluster distances of subclones in T2 tumor. The analysis demonstrates that tumor subclones exhibit a tendency to be spatially close to each other, as indicated by the shorter distances observed within clusters compared to distances between clusters.

Figure 7 was generated by integrating the chromosomal aberration data, including copy number alterations, single nucleotide variations, and structural variations, obtained from our bMDA-seq analysis. We evaluated the presence of shared subclonal mutations among different spatial regions to

infer the evolutionary relationships between the microniches based on their genomic profiles. We apologize for the lack of clarity in our description and will revise the text and methods section to provide a more detailed explanation of the process.

We also performed a deeper spatial analysis in the comment #5.

4b) Selection of spatial samples: The spatial samples were selected by expert pathologists who aimed to capture as many representative heterogeneous subclones as possible. Their selection was based on morphological characteristics, H&E staining, and other pathological features that suggested the presence of distinct tumor subclones. By selecting representative spatial samples, we ensured that our analysis captured the diverse genomic landscape of the tumor and provided insights into the spatial organization of its subclones. To clarify this point, we added how the sample was selected in the methods section in the revised manuscript.

We appreciate your feedback and will incorporate these clarifications and additional analyses into our revised manuscript. This will help to strengthen our study by providing a more comprehensive understanding of the spatial context of tumor evolution and the potential applications of our bMDA-seq method in this area.

5) The authors qualitatively describe many aspects of genomic alterations between spatial samples of TNBC, however, it's not clear any insights or interpretation beyond visualizing them was performed in this manuscript. This may be fine if the authors focus on bMDA technology, but probably does not constitute: "assessed the subclonal evolutionary relationship using integrative genomic analysis within a spatial context" as stated in the abstract.

Thank you for your comment regarding the qualitative description of genomic alterations and the need for a more in-depth interpretation of our findings. We acknowledge that our manuscript could benefit from a more comprehensive analysis of the spatial context and its implications for tumor evolution. To address this, we include spatial insights by visualizing the evolutionary direction using arrows on top of the tissue image (**Figure R11, Fig. 7**).

Specifically, we utilized the diffusion pseudotime algorithm using somatic SNVs and the spatial location information of each spatial microniche. To infer spatial direction of evolution for each specific spatial microniche, a weighted sum of the three spatial vectors pointing from the node prior in pseudotime to the next node later in pseudotime was calculated. The difference in pseudotime between these nodes served as the weight in this calculation. The Pseudo-time trajectory analysis revealed a potential evolutionary direction displayed on top of tissue image, with subclone c1 serving as the ancestral lineage and sequentially giving rise to subclones c2, c3, c4, and ultimately c5.

It is important to note that the tumor section represents a snapshot of the tumor's evolution. Therefore, these subclones are believed to have undergone diversification over time, leading to their observed spatial distribution. Therefore, to visualize evolutionary dynamics and temporal progression, we integrated subclone-specific mutations, including CNAs SNVs, SVs, and kataegis. Overlapping mutations between different subclones was regarded as an evidence for the presence of a common ancestor, suggesting shared evolutionary origins. Subclone-specific mutations were considered as evolutionary changes specific to each subclone over time, reflecting the progressive development and diversification of subclones.

Figure R11 Visualizing the inferred evolutionary direction on top of the tissue image

By adding these visualizations, we aim to provide clinically important insights into tumor development by revealing the spatial direction of mutation development. This approach will help us better understand the migration patterns of tumor subclones and the factors that drive their expansion within the tumor microenvironment. Additionally, these insights can contribute to the development of more targeted therapeutic strategies, as they enable us to identify the subclones with the highest potential for invasion and metastasis.

We believe that by incorporating these spatial insights into our analysis, we will strengthen our claim of assessing the subclonal evolutionary relationship using integrative genomic analysis within a spatial context, as stated in the abstract. We will ensure that these additional analyses and their implications are clearly described in our revised manuscript, providing a more complete understanding of the potential applications of our bMDA-seq method in the field of spatial genomics.

To address this comment, we revised main figure 7 as well as its caption. We also added the method for visualizing spatial analysis in the methods section in the revised manuscript.

6) PTA (<https://www.ncbi.nlm.nih.gov/pmc/articles/PMC8214697/>) is a recently developed alternative to MDA. Can the authors compare to bMDA to PTA? Is a similar barcoding approach also applicable?

Thank you for bringing our attention to the recently developed Primary Template-directed Amplification (PTA) method as an alternative to MDA. We appreciate your suggestion to compare bMDA to barcoded PTA (bPTA) and to explore the possibility of applying a similar barcoding approach to PTA.

Comparison of MDA and PTA:

Multiple Displacement Amplification (MDA) is a widely used method for whole-genome amplification that is based on the use of ϕ 29 DNA polymerase, which has high processivity and strand displacement activity. MDA is known for its high genome coverage and low amplification bias, making it a suitable choice for single-cell and low-input DNA studies. However, MDA can suffer from chimeric DNA formation and amplification artifacts, which could affect the accuracy of downstream analyses.

Primary Template-directed Amplification (PTA) is a more recently developed alternative to MDA. PTA also utilizes ϕ 29 DNA polymerase, but it employs a different amplification strategy. In PTA, the reaction begins with the hybridization of random hexamer primers to the template DNA, followed by extension with ϕ 29 DNA polymerase. This initial step generates a pool of primary amplification products, which serve as templates for subsequent amplification by ϕ 29 DNA polymerase. PTA has been shown to produce high yields of amplified DNA and has a lower rate of chimeric DNA formation compared to MDA.

Both MDA and PTA have their advantages and disadvantages, and the choice between them depends on the specific requirements of the study. MDA is well-established and offers high genome coverage, which is crucial for comprehensive spatial genomics analysis. On the other hand, PTA may provide an alternative approach with reduced chimeric DNA formation, and claims that it has lower amplification bias compared to MDA.

We believe that comparing MDA and PTA is an important aspect of understanding the potential of different amplification methods for spatial genomics studies. Future research may explore the

possibility of developing barcoded PTA (bPTA) to leverage the advantages of both the barcoding approach and PTA. This would further advance the field of spatial and single-cell genomics.

Feasibility of barcoding primers in PTA:

Based on our initial assessment, it seems that a barcoding approach could potentially be applied to PTA by finding a suitable combination of random primers to create a bPTA method. However, there might be challenges in implementing bPTA for SNV analysis, as the amount of short fragment DNA in the PTA product pool is much smaller than that in the MDA product, which could hinder efficient amplification for SNV detection.

Nevertheless, we believe that this is a promising avenue for future research and worth exploring. We appreciate your recommendation and will include a discussion of the potential development of bPTA in the future work section of our manuscript:

“It should also be noted that the use of PTA with cells in tissue sections has not yet been demonstrated, and its suitability for spatial genomics applications and applicability to barcoding strategies remains to be investigated.”

This will highlight the possibility of expanding our barcoding approach to alternative amplification methods and further advance the field of spatial and single-cell genomics.

Minor:

Are H&E images or other images associated with the laser selection available? If so, they should be included with the paper.

We included the image in Supplementary figure 7 (**Figure R12**). Thank you very much for pointing this out.

Figure R12 H&E images associated with the laser isolation

Reviewer #2 (Remarks to the Author):

Dear authors,

Thank you and kudos to successfully developing and present the work of new way of barcoding primer MDA (Multiple Displacement Amplification), that overcomes current barriers of conventional MDA, that limits the adaptation of assay despite inherit benefits.

For the manuscript to benefit more readers and the field, here are some suggestions for authors to consider.

Thank you for your encouraging comments regarding our work on barcoded primer Multiple Displacement Amplification (bMDA). We are pleased to hear that you recognize the potential of this new technique to address the limitations of conventional MDA and to advance the field of genomics.

We appreciate your willingness to provide suggestions for improving our manuscript. Your insights as an expert reviewer will undoubtedly contribute to a more informative and beneficial publication. We look forward to your specific comments and will strive to address each one comprehensively in the next revision of our manuscript.

1. While the bMDA addressed the reduced cost of assay by reducing the costly prepping of the NGS library, it also potentially increases the diversity and representation of the heterogeneous intratumoral population of cells from different regions and potentially indicates where they originated. Would that offer the additional benefit of spatial resolution of the assay? If so, how would that compare to more commonly used technologies like 10x genomics or MerFISH? What would be the major advantage of this approach?

We appreciate your thoughtful query and the opportunity to clarify the significant advantages offered by our bMDA method.

You correctly identified that bMDA may enhance the diversity and representation of heterogeneous intratumoral cell populations across different regions, with the potential to trace their spatial origins. This aspect indeed represents a significant strength of our method, contributing to its potential for high-resolution spatial genomic analysis.

However, it's crucial to differentiate between the types of data provided by various technologies. The referenced 10x Genomics and MerFISH platforms predominantly focus on spatial transcriptomics, offering rich insights into gene expression patterns. Nonetheless, these technologies do not directly supply data about genomic alterations such as copy number alterations (CNAs), single nucleotide variations (SNVs), and Structural Variations (SVs) that cannot be analyzed with transcriptomics.

Conversely, bMDA, developed for spatial DNA sequencing (genomics), effectively detects such genomic alterations. This capability offers a profound understanding of the genetic heterogeneity within tumors, potentially tracing the origin of different cell sub-populations via their unique genetic signatures.

One of the primary goals of our research is to harness this potential to create personalized tumor monitoring DNA panels for patients (**Figure R13**). By mapping the spatial composition of different tumor subclones and their genomic aberrations, we aim to offer clinicians a powerful tool for personalized diagnostics and treatment planning. This cannot be accomplished by more commonly used technologies like 10X genomics or MerFISH.

An example of usage of spatial genomics data to building bespoke ctDNA panel

Figure R13. An example of usage of spatial genomics to build bespoke ctDNA panel.

In summary, bMDA's primary advantage lies in its ability to deliver spatially resolved genomic data. This complements existing spatial transcriptomic methods, promoting a more thorough understanding of intra-tumoral heterogeneity.

2. Given the success and importance of immune-oncology in breast cancer, would the authors consider analyzing the immune cell population using this technology? This can be informative – especially one can be informed about the spatial origin of the cells, and this can inform the benefit of checkpoint inhibitors and other immune-related therapeutics

We greatly value your suggestion to extend the application of bMDA technology to analyzing immune cell populations, given the critical role of immuno-oncology in breast cancer treatment. Indeed, understanding the spatial origin of immune cells within the tumor microenvironment has significant potential for informing therapeutic strategies, including the use of checkpoint inhibitors.

Figure R14. Reconstituted figure from our previous work: Lee et al. Nat. Comm. 2022. The figure describes SLACS technology combined with full-length RNA sequencing, which is more adequate for delineating immune cell population.

In previous research published in Nature Communications (Lee et al., *Nature Communications*, 2022, **Figure R14**), we successfully utilized a spatial transcriptomics platform to delineate the immune cell population, and we appreciate the implications of this approach for immuno-oncology.

However, it is important to note the primary application of bMDA is for DNA sequencing, which is particularly relevant for analyzing genomic aberrations in cancer cells. Tumor cells typically possess a multitude of genomic alterations, and bMDA offers a powerful tool for detecting and mapping these aberrations in a spatial context. On the other hand, immune cells within the tumor microenvironment, while of great interest in cancer research and treatment, typically do not exhibit the same degree of genomic alterations. Hence, the capacity of bMDA to detect genomic aberrations might not be fully exploited in the context of immune cell analysis.

Our bMDA-seq target selection process balanced two crucial considerations: the representation of heterogeneous subclones and the minimization of healthy cell contamination. Our expert pathologists used morphological characteristics, H&E staining, and other pathological indicators to select areas with distinct tumor subclones and a high tumor cell density. This focus on tumor-enriched regions allowed us to maintain high tumor purity in our selected samples, achieving 98.8% purity compared to 51% in bulk genomic DNA (**Figure R15, Supplementary Fig. 8 a,b**). We have elaborated on this selection process in the methods section of our revised manuscript to provide further clarity.

Figure R15. Estimated tumor purity of tumor bulk and the isolated cell clusters analyzed by bMDA-seq. The analysis reveals negligible healthy cell contamination in the cell clusters. Tumor purity was estimated using the ABSOLUTE algorithm based on the allelic frequency of somatic SNVs.

That said, we do acknowledge the critical role of immune cells in the tumor microenvironment, and we're planning to incorporate pathological image analysis in our future studies to offer a more comprehensive insight into the immune cell population in the context of the tumor spatial heterogeneity.

In summary, while bMDA could technically be used to analyze any cell population, its unique capabilities are particularly well-suited for investigating the genomic landscape of cancer cells. We will, however, continue to explore its potential applications in various research areas including pathological image analysis to provide a broader perspective on tumor heterogeneity.

3. bMDA has shown to detect the structural variation and kataegis effectively – would this technology be applied in detecting germline mutation, with lower cost and higher efficacy potentially? Is there a reason to believe this can potentially detect currently undetectable germline mutation if we were to use normal cells?

Thank you for your insightful comment. Indeed, the application of bMDA in detecting germline mutations and studying somatic mosaicism presents an exciting avenue for future research. The high coverage and sensitivity of bMDA for detecting structural variations and kataegis suggest that it could be highly effective in identifying germline mutations, potentially at a lower cost and with higher efficiency compared to existing methods.

bMDA's true potential can be harnessed in scenarios where the normal cells in a sample are heterogeneous and carry different germline mutations; thereby requiring whole genome amplification. In such cases, bMDA could provide a higher resolution snapshot of the genomic landscape and elucidate intricate details of the genetic variations at a single-cell level, a feat difficult to achieve through bulk sequencing.

In contrast, if the normal cells all carry the same germline mutations, bulk sequencing might be a more suitable approach, considering cost-effectiveness and simplicity. However, the bMDA approach would still offer an advantage in detecting mosaicism and capturing the full spectrum of genomic diversity within the sample.

Furthermore, in the context of germline mutations, bMDA could be a powerful tool for identifying variants associated with hereditary diseases, thereby aiding early diagnosis and intervention; in cases where minute amount of genomes can be used. bMDA could also uncover currently undetectable germline mutations if applied to normal cells, with significant implications for studying somatic mosaicism, a field of growing interest in human genetics.

We greatly appreciate your suggestion and plan to explore the potential of bMDA in the detection of germline mutations and the study of somatic mosaicism in our future research. Your feedback is invaluable to our ongoing efforts to develop and refine this technology.

To address this comment we added sentences to the discussion section regarding future use of bMDA-seq.

4. In the same token as #3, can we use this assay to detect the potential benefit of DNA-damage repair pathway targeted therapeutics by genomic event recognition, which is not possible by bulk sequencing?

We appreciate your suggestion, and we agree that the unique capabilities of bMDA could be leveraged to assess the potential benefit of DNA-damage repair pathway-targeted therapeutics. Specifically, the high-coverage and sensitivity of bMDA could enable the detection of specific genomic events, such as structural variants, single nucleotide variants, and patterns of kataegis, which are indicative of defective DNA-damage repair pathways.

One of the challenges with bulk sequencing is the dilution of signals from rare or minority populations of cells, which can cause potentially crucial genomic events to be overlooked. This problem can be circumvented by the application of bMDA, which can isolate and amplify the genomic content of individual cells, thus preserving the integrity of such signals.

Furthermore, bMDA could potentially be used to map the spatial distribution of these genomic events within a tumor, providing insights into the intratumoral heterogeneity of DNA repair defects. This could further inform the choice of DNA-damage repair pathway-targeted therapeutics, potentially guiding personalized treatment strategies.

Indeed, a promising application of this technology could be in the context of cancers with BRCA1/2 mutations, which are known to have defective homologous recombination DNA repair pathways and are sensitive to PARP inhibitors.

By applying bMDA, we were able to identify subclonal populations of cells carrying deleterious protein-altering mutations that were not detected by bulk sequencing. These mutations hold potential as therapeutic targets and could be considered for targeted therapies. However, further studies are necessary to investigate the functional impact of these mutations, their associations with treatment response or prognosis, and their suitability as therapeutic targets in breast cancer.

Specifically, we observed the presence of the following genes in distinct clusters:

BCL11A (cluster c1, c3): Inhibiting *BCL11A* or its downstream pathways is being explored as a potential treatment strategy, particularly in hematological malignancies.

TTK (cluster c3): *TTK* inhibitors have been investigated as potential anticancer agents, particularly in tumors with high mitotic activity.

MARK4 (cluster c1, c2, c4, c5): Inhibition of *MARK4* or related pathways is being explored as a potential treatment option, particularly in prostate cancer.

ADRA1B (cluster c3): This gene encodes a receptor involved in cell signaling pathways and has been associated with various cancers, including breast cancer. It may have implications for treatment response and warrants further investigation.

Thank you for your insightful comment. We will consider exploring these applications in our future research with bMDA.

To address this comment we added sentences to the discussion section regarding potential use of bMDA-seq. We also added full list of detected mutations in the **Supplementary Data**.

5. It seems though – even if the bMDA may be better generalizable, one should still understand the MDA (conventional) assay itself to use the bMDA to its maximum benefit. For instance, to reduce the barcode bias – and also juxtapose the non-biased amplification of gene loci, false positive correction etc. How would one envision that this knowledge can be successfully transferred to other labs?

Thank you for raising this important point. While we have made efforts to streamline and simplify the bMDA process, a fundamental understanding of the principles and mechanics of the conventional MDA is indeed useful for the successful application of bMDA.

To aid in the transfer of this knowledge to other labs, we have provided a detailed protocol in the methods section of our manuscript. In addition, we plan to publish a more comprehensive step-by-step protocol to guide researchers in carrying out bMDA. This will include explicit instructions on how to mitigate barcode bias and perform false positive correction, among other things.

Further, we intend to make ourselves available for any questions or clarifications that other researchers may have when implementing bMDA in their labs. Through these measures, we hope to ensure that the bMDA method can be broadly adopted and effectively utilized.

In summary, we believe that through careful guidance and open communication, the knowledge required to successfully use bMDA can be effectively transferred to other labs. We are committed to supporting the scientific community in adopting this new technology.

To address this comment, the full method will be available via the manuscript and supplementary information.

6. Given the advantage of this technology in detecting the microniche and phylogeny of heterogeneous tumors, would authors consider comparing primary vs metastatic tumor or primary vs lymph node, or even pre-cancerous lesion vs cancer from the same patients? This can rationalize the additional benefit /utility of this novel assay.

We greatly appreciate your suggestion and completely agree that applying the bMDA approach to the scenarios you described would provide invaluable insights into tumor evolution and heterogeneity. Comparing primary tumors to metastatic sites or lymph nodes, or pre-cancerous lesions to fully developed cancer, could unravel the genomic underpinnings of disease progression and metastasis, and highlight key differences in the mutational landscapes and subclonal compositions.

Moreover, comparing genomic data from different tumor sites and stages of disease progression within the same patient could offer personalized insights, potentially informing more targeted and effective treatment strategies for individual patients. This could indeed underline the utility of the bMDA approach in a clinical setting.

As such, we plan to expand the application of our bMDA technology in future studies to include comparisons between primary and metastatic tumors, lymph nodes, and pre-cancerous lesions from the same patients. We also have previous study that has applied MDA to circulating tumor cells (CTCs) and are planning to apply bMDA to CTCs for effective and precise genomic abnormality detection (Kim et al. Small, 2019). These studies would provide a more comprehensive understanding of tumor heterogeneity and evolution, demonstrating the additional benefits and potential clinical applications of our novel bMDA approach. Your suggestion will be added to the discussion section as a part of future work directions.

7. While the manuscript is overall well-written, it can also benefit from proofreading. Some sentences in the results can also be moved to either discussion or methods section.

Thank you for the valuable feedback. We appreciate your observation regarding the placement and clarity of certain sentences within the manuscript. We will diligently review the manuscript to further improve its clarity and readability. In particular, we will reassess our results section to ensure that it strictly presents our findings and move any technical or interpretative details to the appropriate methods or discussion sections. We will also ensure a thorough proofreading of the entire manuscript to eliminate any grammatical errors and enhance its overall coherence. We strive to present our work in a clear, succinct, and reader-friendly manner, and your feedback greatly helps us in this endeavor. Thank you once again for your constructive suggestions.

Reviewer #3 (Remarks to the Author):

This study presents the development and application of barcoded multiple displacement amplification (bMDA) for scalable and comprehensive genome analyses. The incorporation of cell barcodes into MDA products using barcoded primers enables sample pooling and streamlines library preparation. The study successfully demonstrated the preparation of 720 bMDA libraries in only 15 tubes with high performance comparable to conventional MDA, achieving a 48-multiplexed sequencing library per reaction tube. The single-cell bMDA data exhibited sufficient genetic coverage for genome analyses at single-nucleotide resolution.

Given that the introduction of the barcoded primer to MDA is the main technical advancement in this study, the authors present extensive data showing the the amplification efficiency of MDA with different barcoding strategies and found that the increased length of the barcoded primer with the unusual high concentration of the primer in MDA inhibited the bMDA reaction. To achieve the best coverage and multiplexing ability of bMDA, they choose 6-mer cell barcode primer with the final proportion 2% out of total MDA N6 primers.

This paper is organized and logically structured, presenting the development and application of bMDA for scalable genome analysis. It addresses technical challenges and emphasizes the potential of bMDA in spatial genomics with the analysis of a breast cancer sample provided as an example.

I think this approach has the potential to have a significant impact on the field, and just have a minor suggestions for improving the quality of the manuscript.

We are deeply appreciative of the time you spent reviewing our manuscript and providing your valuable feedback. Your comments reaffirm the value of our work in introducing barcoded multiple displacement amplification (bMDA) for scalable and comprehensive genome analysis. We are glad that you found the study well-organized, logically structured, and impactful for the field.

We eagerly look forward to receiving your minor suggestions to improve the quality of our manuscript. We are committed to addressing these points to further enhance the clarity and potential impact of our work.

1) The Lorenz curve in figure 3e should go to 0 on the x-axis for portions of the genome that were not covered.

Thank you for your suggestion regarding the adjustment of the Lorenz curve in Figure 3e. We concur that the curve should indeed touch zero on the x-axis for portions of the genome that were not covered. As correctly noted by the reviewer, the Lorenz curve is typically used with high depth whole genome sequencing data to visualize read depth bias at base-pair (bp) resolution.

In our study, however, we specifically aimed to evaluate the efficacy of bMDA in detecting copy number alterations (CNAs) through the use of low depth whole genome sequencing data, while simultaneously utilizing high depth targeted sequencing for the detection of SNVs. This combination makes bMDA particularly effective and powerful in analyzing comprehensive genetic alterations.

Although our approach using fractions of bins, instead of fractions of genome (bp), may not provide a traditional representation of amplification bias, we have previously demonstrated (S. Kim et al., *Genome Biology*, 2018) that a Lorenz curve calculated by fraction of bins can still be a valuable measure of amplification bias (**Figure R16**).

We appreciate your feedback and hope this clarification addresses your concerns. In light of the reviewer's comments, we have added to and revised our methods and figure caption to clarify this approach.

Figure R16 In our QC process, we utilized the Amplification Start Time (AST) of the real-time MDA amplification reaction. (c) The correlation between the AST value and the area under the Lorenz curve indicates that the AST value can serve as a reliable estimation of amplification uniformity. (d, e) The high correlation of CNAs between samples with a low AST value suggests that the Lorenz curve calculated by the fraction of bins can be a valuable measure of amplification bias.

2) For figure 4g, what was the SNV calling specificity/precision?

We apologize for the oversight in the initial manuscript that did not specifically mention this information, and we acknowledge the importance of providing these details to assess the reliability of our bMDA sequencing method.

The precision and specificity of SNV calling are described in Figure 4b and Supplementary Figure 8d (**Figure R17**). However, we realize that these descriptions were not adequately explained in the manuscript. In the revised version, we will provide a comprehensive and detailed explanation of our approach to SNV calling in the Methods section and figure caption, addressing the precision and specificity metrics. This will ensure that readers have a clear understanding of the reliability and accuracy of our SNV calling methodology.

Figure R17 SNV calling sensitivity and specificity for cell clusters of T1 and T2 TNBC tumor

Thank you again for highlighting this important point. We appreciate your careful attention to detail, which is helping us to improve the clarity and rigor of our study.

To address this comment, we corrected the figure caption.

3) For all the types of genetic variation, what were the authors able to detect in multiple samples that were missed with bulk sequencing? Were any targetable lesions missed that could have changed the patient's treatment?

Thank you for your thoughtful comments and questions. In response to your third point, we acknowledge the potential of our barcoded multiple displacement amplification (bMDA) approach in uncovering novel insights into tumor heterogeneity that were previously missed by bulk sequencing.

Indeed, our bMDA method was able to detect diverse genomic aberrations, such as subclone-specific single nucleotide variations (SNVs), kataegis, and structural variations (SVs), which are often overlooked in bulk sequencing due to the 'averaging out' effect. Moreover, the high-resolution nature of our method enables the precise delineation of copy number alterations (CNAs) without the confounding 'averaging' issues inherent to bulk sequencing.

You have astutely pointed out the potential clinical relevance of this ability to detect subclonal SNVs. The identification of these genomic variations could indeed lead to the discovery of clinically actionable somatic mutations that could potentially guide personalized therapeutic strategies. Therefore, our bMDA approach could directly inform treatment plans and could potentially improve patient outcomes by enabling a more tailored therapeutic approach.

By applying bMDA, we were able to identify subclonal populations of cells carrying protein-altering mutations that were not detected by bulk sequencing (**Supplementary Data**). These mutations hold potential as therapeutic targets and could be considered for targeted therapies.

Our study revealed multiple instances of genetic variations missed by bulk sequencing, some of which could potentially influence therapeutic decisions.

Firstly, although not classified as actionable, variations in the *ESR1* gene often guide the use of Elacestrant. Notably, we identified some subclones in the T2 tumor exhibiting an *ESR1* p.R548H mutation, which was missed in the bulk sequencing.

Moreover, according to the OncoKBTM database, the amplification of the *CCNE1* gene is associated with responsiveness to therapies like RP-6306 and BLU-222. While bulk sequencing of the T1 tumor showed no *CCNE1* gene aberrations, our bMDA-seq detected subclones with amplified *CCNE1* genes. Interestingly, we observed both amplification and deletion of the *CCNE1* gene in subclones of the T2 tumor, despite the bulk population primarily displaying amplification. These findings highlight the significance of recognizing such heterogeneity within subclonal populations when contemplating therapeutic strategies.

As the field evolves, with an increasing number of mutation-matched therapeutic options being developed, we anticipate that tools like bMDA-seq will play a crucial role in deciphering the most effective treatment options by illuminating the intricate landscape of genetic variations within tumors.

Moreover, our approach could contribute significantly to the development of personalized tumor relapse monitoring panels. By detailing the spatial composition of different subclones and their genomic aberrations, we aim to create specific panels that could be used for the detection of circulating tumor DNA (ctDNA) in patients who have undergone surgery. This approach could potentially enable early detection of relapse, thus providing an opportunity for prompt intervention. One of the primary goals of our research is to harness this potential to create personalized tumor monitoring DNA panels for patients (**Figure R18**). By mapping the spatial composition of different tumor subclones and their genomic aberrations, we aim to offer clinicians a powerful tool for personalized diagnostics and treatment planning. This cannot be accomplished by more commonly used technologies like 10X genomics or MerFISH.

Figure R18. An example of usage of spatial genomics to build bespoke ctDNA panel.

We appreciate your insightful suggestions and will ensure to emphasize these points more clearly in our revised manuscript. We also added full list of detected mutations in the **Supplementary Data**. Your recommendations have been invaluable in helping us articulate the potential impact and applications of our work.

REVIEWERS' COMMENTS

Reviewer #1 (Remarks to the Author):

The authors have done a commendable job of addressing my comments.

I have one additional suggestion, I believe the diffusion pseudotime visualization in F11, Fig 7 might be speculative, due to cell migration, and local tissue remodeling. I would recommend removing this analysis.

Reviewer #2 (Remarks to the Author):

I thank the authors for addressing the concerns that reviewers have brought. I am happy with the revisions that are made/submitted and propose that we grant publication of this revised manuscript.

Reviewer #3 (Remarks to the Author):

The authors have adequately addressed my concerns-- I believe that the manuscript is now suitable for publication in Nature Communications.

Point-by-point response

Reviewer #1 (Remarks to the Author):

The authors have done a commendable job of addressing my comments.

I have one additional suggestion, I believe the diffusion pseudotime visualization in F11, Fig 7 might be speculative, due to cell migration, and local tissue remodeling. I would recommend removing this analysis.

We would like to express our sincere gratitude for recognizing the efforts we have made in addressing your previous comments and for providing further constructive feedback on our manuscript.

Your additional suggestion regarding the diffusion pseudotime visualization in F11, Fig 7 has been carefully considered. We understand your concerns about the potential speculative nature of this analysis due to cell migration and local tissue remodeling.

After extensive deliberation, we have decided to comply with your recommendation and have removed the diffusion pseudotime visualization from Fig 7 in our manuscript. We believe that this alteration helps to enhance the integrity and robustness of our research, and we appreciate your expert insight in guiding this decision.

We revised the part where Fig 7 is explained in the main manuscript as follows:

“Furthermore, by integrating all chromosomal aberrations (Supplementary Data), we constructed a spatial map depicting the inferred evolutionary relationships between these microniches (Fig. 7). While subclone c1 seems to serve as the ancestral lineage and sequentially gives rise to subclones c2, c3, c4, and, ultimately, c5, it is important to emphasize that the tumor section represents a snapshot of the tumor's evolution. These subclones are believed to have undergone diversification over time (Fig. 7), leading to their observed spatial distribution. Rather than positing a specific evolutionary direction, this

analysis provides valuable insights into the spatial relationships and potential evolutionary dynamics of tumor subclones within tissues.”

Once again, thank you for your constructive and thoughtful comments. We look forward to hearing any additional feedback you may have and are committed to ensuring that our work meets the high standards of Nature Communications.

Reviewer #2 (Remarks to the Author):

I thank the authors for addressing the concerns that reviewers have brought. I am happy with the revisions that are made/submitted and propose that we grant publication of this revised manuscript.

We wholeheartedly thank you for your kind words and for recognizing the revisions we made to our manuscript in response to the concerns that you and other reviewers raised.

Your satisfaction with our efforts and your proposal to grant publication of this revised manuscript are deeply encouraging to our team. We are grateful for your expert guidance and support throughout this process, and we feel honored to have our work considered for publication in Nature Communications.

Once again, thank you for your invaluable feedback and encouragement. We look forward to the opportunity to contribute our research to the scientific community through Nature Communications.

Reviewer #3 (Remarks to the Author):

The authors have adequately addressed my concerns-- I believe that the manuscript is now suitable for publication in Nature Communications.

We extend our heartfelt thanks for acknowledging our efforts to address your concerns and for finding our manuscript suitable for publication in Nature Communications.

Your expert insights and feedback have played an instrumental role in shaping and improving our work. We greatly value your support and the time you invested in the review process.

With your endorsement, we eagerly anticipate the opportunity to share our research with the broader scientific community through Nature Communications. We are grateful for your positive assessment and encouragement.

Once again, thank you for your invaluable contribution to our work.